# Conduction pathway for potassium through the *Escherichia coli* pump KdpFABC

**Adel Hussein[1], Xihui Zhang[1], Bjørn P Pedersen[2], David L Stokes[1]\***

[1]Department of Biochemistry and Molecular Pharmacology, NYU School of Medicine, New York, United States; [2]Department of Molecular Biology and Genetics, Aarhus University, Aarhus, Denmark

---

## eLife Assessment

This article revisits the well-studied KdpFABC potassium transport system from bacteria with a **convincing** set of new higher resolution structures, a protein expression strategy that permits purification of the active wildtype protein, and insight obtained from mutagenesis and activity assays. The thorough and thoughtful mechanistic analyses make this a **valuable** contribution to the membrane transport field.

---

**\*For correspondence:**
stokes@nyu.edu

**Competing interest:** The authors declare that no competing interests exist.

**Abstract** Under osmotic stress, bacteria express a heterotetrameric protein complex, KdpFABC, which functions as an ATP-dependent $K^+$ pump to maintain intracellular potassium levels. The subunit KdpA belongs to the superfamily of $K^+$ transporters and adopts pseudo fourfold symmetry with a membrane-embedded selectivity filter as seen in $K^+$ channels. KdpB belongs to the superfamily of P-type ATPases with a conserved binding site for ions within the membrane domain and three cytoplasmic domains that orchestrate ATP hydrolysis via an aspartyl phosphate intermediate. Previous work hypothesized that $K^+$ moves parallel to the membrane plane through a 40 Å long tunnel that connects the selectivity filter of KdpA with a canonical binding site in KdpB. In the current work, we have reconstituted KdpFABC into lipid nanodiscs and used cryo-EM to image the wild-type pump under turnover conditions. We present a 2.1 Å structure of the E1~P·ADP conformation, which reveals new features of the conduction pathway. This map shows strong densities within the selectivity filter and at the canonical binding site, consistent with $K^+$ bound at each of these sites in this conformation. Many water molecules occupy a vestibule and the proximal end of the tunnel, which becomes markedly hydrophobic and dewetted at the subunit interface. We go on to use ATPase and ion transport assays to assess effects of numerous mutations along this proposed conduction pathway. The results confirm that $K^+$ ions pass through the tunnel and support the existence of a low-affinity site in KdpB for releasing these ions to the cytoplasm. Taken together, these data shed new light on the unique partnership between a transmembrane channel and an ATP-driven pump in maintaining the large electrochemical $K^+$ gradient essential for bacterial survival.

## Introduction

The Kdp ('K-dependent') system was discovered in the 1970s as one of three major systems for maintaining intracellular $K^+$ levels in *Escherichia coli* (*Epstein and Davies, 1970*; *Rhoads et al., 1976*). The transmembrane gradient of $K^+$ maintained by these systems represents a fundamental energy source that enables a variety of secondary processes necessary for cell growth and homeostasis (*Stumpe et al., 1996*). Further work revealed that the Kdp system consists of two overlapping operons with six

genes comprising kdpFABC and kdpDE. The former encodes a hetero-tetrameric pump that binds K$^+$ with high affinity and uses ATP to transport it into the cell (*Gassel et al., 1999*; *Laimins et al., 1978*). The latter encodes a two-component regulatory system in which KdpD senses osmotic stress (e.g., low extracellular K$^+$ concentrations) and acts as a histidine kinase to activate KdpE, which in turn induces expression of kdpFABC (*Walderhaug et al., 1992*). More recently, KdpD was discovered to have a second function as a serine kinase that acts directly on the KdpFABC protein complex to inhibit its activity when normal K$^+$ levels have been restored in the growth media (*Silberberg et al., 2024*).

KdpA and KdpB subunits descend from two distinct and well-studied membrane transport super-families and, in KdpFABC, engage in an unprecedented partnership to actively pump K$^+$ into the cell (*Pedersen et al., 2019*). Early studies identified a series of mutants in KdpA affecting the apparent affinity of the Kdp system, thus determining the minimal K$^+$ concentration at which cells would grow (*Buurman et al., 1995*; *Epstein et al., 1978*). Later, it was recognized that KdpA belongs to the Superfamily of K$^+$ Transporters (SKT superfamily), which also includes bona fide K$^+$ channels such as TrkH and KtrB (*Diskowski et al., 2015*; *Durell et al., 2000*). As such, KdpA is characterized by pseudo fourfold symmetry and a 'selectivity filter' that captures ions from the periplasm (*Miller, 2000*). KdpB was recognized to have signature sequences of the P-type ATPase superfamily that are generally known for coupling ATP hydrolysis to transport of substrates across membranes of organisms from all kingdoms of life (*Hesse et al., 1984*). This association was confirmed by its formation of an aspartyl phosphate intermediate and characteristic inhibition by vanadate (*Puppe et al., 1992*). Function and phylogenetic relationships of KdpF and KdpC are less clear, though both are suggested to serve as structural chaperones that stabilize the hetero-tetrameric complex (*Pedersen et al., 2019*). Like other P-type ATPases, KdpFABC employs the Post-Albers reaction cycle (*Figure 1*) involving two main conformations (E1 and E2) and their phosphorylated states (E1~P and E2-P) to drive transport (*Albers, 1967*; *Post et al., 1969*).

A series of X-ray and cryo-EM structures of KdpFABC from *E. coli* (*Huang et al., 2017*; *Silberberg et al., 2022*; *Silberberg et al., 2021*; *Stock et al., 2018*; *Sweet et al., 2021*) indicate a novel trans-port mechanism befitting the unprecedented partnership of these two superfamilies within a single protein complex. As first proposed by *Stock et al., 2018*, there is now a consensus that K$^+$ enters the complex from the extracellular side of the membrane through the selectivity filter of KdpA, but is blocked from crossing the membrane. Instead, it travels parallel to the membrane plane through a 40 Å long tunnel to reach KdpB. Here, K$^+$ moves to a binding site in the transmembrane domain of KdpB that resembles canonical sites from other P-type ATPases. Like these other pumps, binding of ions triggers autophosphorylation of a conserved aspartate (Asp307) in the phosphorylation domain (P-domain). Consequent conformational changes in KdpB pinch off the tunnel and induce release of K$^+$ to the cytoplasm, though the precise location of the low-affinity, release site remains uncertain. In addition, there is uncertainty about the content of the tunnel which has been modeled either as water (*Sweet et al., 2021*) or dehydrated K$^+$ ions (*Silberberg et al., 2022*; *Silberberg et al., 2021*) in previous work. To shed further light on this unusual transport mechanism, we employ functional and structural approaches to characterize the conduction pathway of K$^+$. We mutated a series of residues along this pathway to assess their functional impact using both ATPase and transport activities. We also determined a cryo-EM structure at substantially higher resolution (2.1 Å) compared to previous structural studies. In addition, the structure was obtained under the most native conditions to date, namely with uninhibited, wild-type (WT) protein embedded in lipid bilayer nanodiscs under active turnover conditions induced by the addition of K$^+$ and ATP. This structure reveals new details along the transport pathway, including clear evidence of tunnel hydration as well as K$^+$ occupancy in the selec-tivity filter in KdpA and at the canonical binding site in KdpB. Map densities in the catalytic domains of KdpB unambiguously show phosphorylation of Asp307 in the P-domain and binding of ADP together with two Mg$^{2+}$ ions in the nucleotide binding domain (N-domain), confirming the conformation to be E1~P·ADP. Effects of mutation at the entrance of the selectivity filter and within the tunnel confirm the transport pathway. In addition, mutations at the canonical ion site in KdpB as well as at a proposed release site provide experimental evidence for the allosteric changes that drive transport.

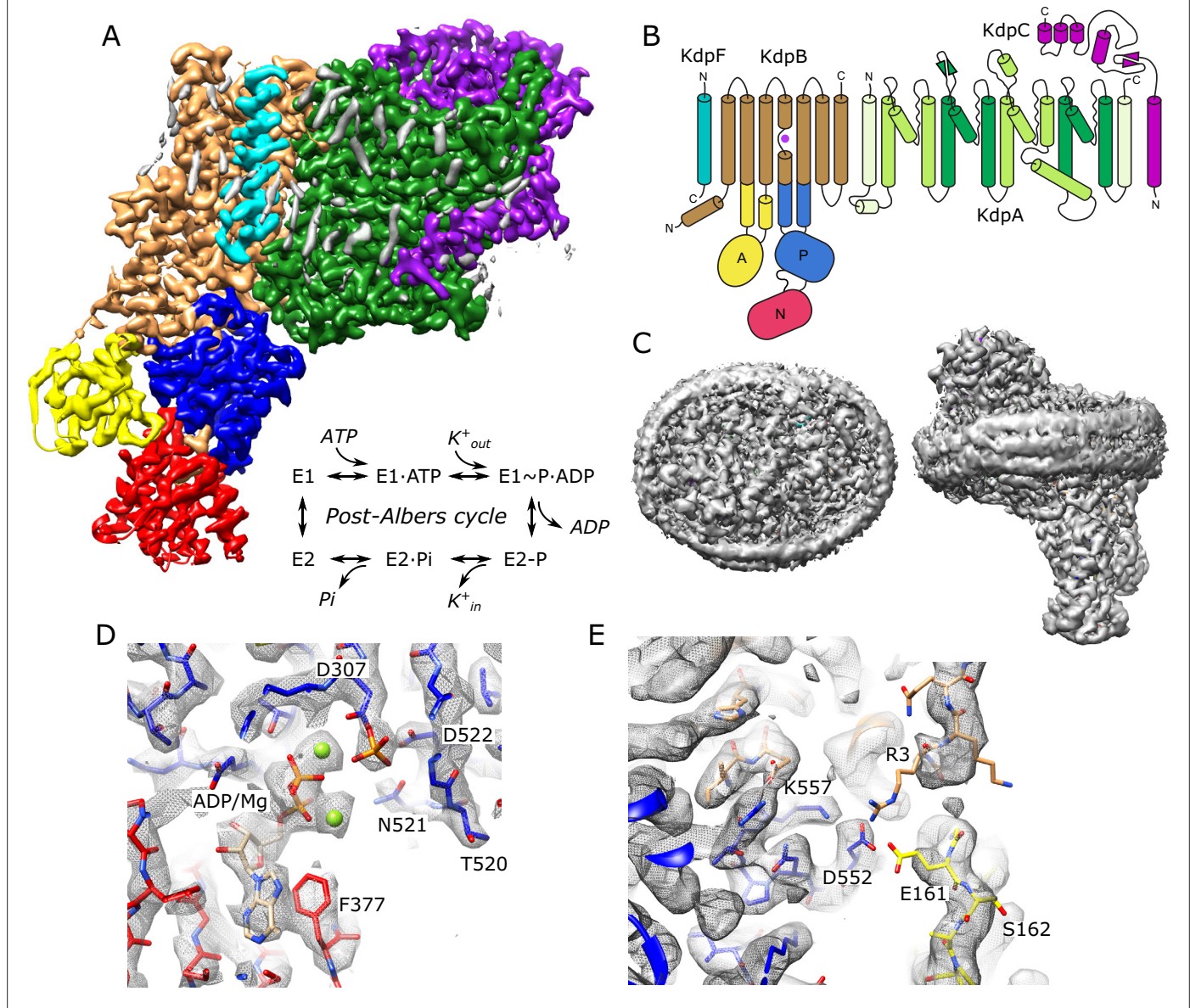

**Figure 1.** Cryo-EM structure of KdpFABC in nanodiscs. (**A**) Overview of the density map with subunits rendered in conventional colors corresponding to those in panel (**B**). The Post-Albers reaction scheme is shown in the inset. (**B**) Diagram illustrating the transmembrane topology of KdpFABC. (**C**) Rendering of the unsharpened map at a lower density threshold illustrates the presence of the membrane scaffolding protein surrounding the nanodisc. (**D**) Detail of the nucleotide binding site of KdpB showing ADP associated with two Mg$^{2+}$ ions and the phosphorylated catalytic residue: Asp307. (**E**) Detail of the newly resolved N-terminus of KdpB revealing a potential interaction between Glu161, Arg3, Asp552, and Lys557. Panels (**A**), (**C**), and (**E**) show the unsharpened map at thresholds of 7, 3, and 4 σ, respectively; panel (**D**) shows the sharpened map at 3.9 σ.

The online version of this article includes the following source data and figure supplement(s) for figure 1:

**Figure supplement 1.** Biochemical preparation of KdpFABC.

**Figure supplement 1—source data 1.** Source data for *Figure 1—figure supplement 1B–H*, which includes all individual data points used for these plots.

**Figure supplement 1—source data 2.** Uncropped gel shown in *Figure 1—figure supplement 1a*.

**Figure supplement 1—source data 3.** Uncropped gel shown in *Figure 1—figure supplement 1a* with annotations.

**Figure supplement 2.** Cryo-EM processing pipeline.

**Figure supplement 2—source data 1.** Source data for the Fourier shell coefficient plot in *Figure 1—figure supplement 2*.

**Figure supplement 3.** Cryo-EM densities.

## Results

### Structure of KdpFABC in the E1~P conformation

For expression, we used an *E. coli* strain lacking the dual-function kinase KdpD that is responsible for post-translational inhibition of KdpFABC depending on the growth conditions (*Silberberg et al., 2024*). In particular, phosphorylation of Ser162 by KdpD confounded previous attempts to survey mutations along the putative transport pathway (*Sweet et al., 2020*) and produces off-cycle, inhibitory conformations (*Huang et al., 2017*; *Silberberg et al., 2022*). Use of the KdpD knockout strain allowed us to produce WT and mutant protein free from Ser162 phosphorylation. Our WT complex displayed high levels of K$^+$-dependent ATPase activity and generated robust transport currents (*Figure 1—figure supplement 1*). Unlike previous structural analyses, which relied on detergent-solubilized preparations, we reconstituted the purified complex into lipid nanodiscs to provide a more native lipid environment. Rather than use inhibitors to stabilize a particular state for cryo-EM imaging, we added ATP and froze the sample under active turnover. We collected a large data set (~50,000 micrographs) and used 3D classification to segregate various conformational states (*Figure 1—figure supplement 2*). In this paper, we report on the most highly populated state, which resulted in the highest resolution reported for KdpFABC to date (2.1 Å, *Table 1*) in which all transmembrane helices and cytoplasmic domains are well resolved (*Figure 1—figure supplement 3*). *Figure 1* shows an overview of the structure, including the belt formed by the nanodisc scaffolding protein which is visible at a lower density threshold. Densities at the catalytic site indicate that Asp307 in the P-domain is phosphorylated and that ADP is bound within the N-domain (*Figure 1D*), consistent with the E1~P·ADP conformation. Two Mg$^{2+}$ ions are associated with ADP, consistent with a Hill coefficient of 1.8 determined by Mg$^{2+}$ titration of ATPase activity (*Figure 1—figure supplement 1G*). The overall conformation is quite similar to previous structures of the E1 state in detergent (RMSD for Cα atoms of 0.8–1.5 Å) including E1~P states captured under turnover conditions (*Silberberg et al., 2022*) and the E1·ATP state stabilized with AMPPCP (*Sweet et al., 2021*).

Unlike previous structures, our map resolves most of the N-terminus of KdpB, revealing a potential interaction involving Glu161 from the A-domain, Arg3 from the N-terminus as well as Asp552 and Lys557 from the P-domain (*Figure 1e*). This interaction is notable given the role of Glu161 in hydrolysis of the aspartyl phosphate in the subsequent E2-P state (*Dyla et al., 2020*). Furthermore, post-translational regulation of KdpFABC involves phosphorylation of the neighboring residue, Ser162, which leads to inhibition of this dephosphorylation step and thus cessation of transport (*Sweet et al., 2020*). Given the regulatory roles played by N- and C-termini of a variety of other P-type ATPases (*Bitter et al., 2022*; *Calì et al., 2017*; *Lev et al., 2023*; *Timcenko et al., 2019*; *Zhao et al., 2021*), we generated a construct in which residues 2–9 of the N-terminus of KdpB were truncated. However, ATPase and transport activities remained coupled at levels similar to WT, indicating that any functional role of the N-terminus is relatively subtle and not manifested under current assay conditions.

### Passage of K$^+$ through the selectivity filter of KdpA

Selectivity filters are a defining feature of K$^+$ channels and the SKT superfamily that includes KdpA. The selectivity filter from KcsA has four binding sites denoted S1-S4 (*Morais-Cabral et al., 2001*) and previous structures of KdpFABC have consistently shown strong density only at the equivalent S3 site. In the current map, however, clear densities are visible at all four sites, likely due to increased resolution (*Figure 2A*), but density at S3 remains much stronger. In fact, this S3 site contains the strongest densities in the entire map, measuring 7.9× higher than the threshold used for *Figure 2A* (*Figure 2—figure supplement 1A*). This suggests that S3 is occupied by a well-ordered K$^+$ ion. The higher resolution also allows accurate placement of carbonyl atoms in the selectivity filter and the resulting coordination geometry (*Figure 2—figure supplement 1B*) is consistent with S3 acting as the primary binding site for K$^+$. S3 is tightly coordinated by the eight surrounding carbonyl atoms at an average distance of 2.8 Å (s.d. 0.2 Å), which is consistent with the expected center-to-center distance for K$^+$ and oxygen atoms of its hydration shell (*Naranjo et al., 2016*). This coordination geometry is also consistent with that seen in the K$^+$ channel KcsA, though the strict four-fold symmetry of that homo-tetramer produces a more regular structure, as indicated by the smaller variance in liganding distance (2.77 Å with s.d. 0.075 Å in 1K4C) and as depicted by Huang et al. in Extended Data (*Figure 3*; *Huang et al., 2017*). At S2, coordination is noticeably looser with the four lower carbonyls at an average of 3.0 Å (s.d. 0.9 Å) and the four upper carbonyls at 3.6 Å (s.d. 0.1 Å). The S1 position is only coordinated from

**Table 1.** Structure determination of KdpFABC.

| Deposition | |
| --- | --- |
| PDB | 9OC4 |
| EMDB | EMD-70308 |
| **Data collection and processing** | |
| Magnification | 130kx |
| Voltage (kV) | 300 |
| Electron exposure (e⁻/Å²) | 50 |
| Defocus range (µm) | 1.0–3.0 |
| Pixel size (Å) | 0.93 |
| Symmetry imposed | C1 |
| Initial particle images (no.) | 10,382,000 |
| Final particle images (no.) | 672,405 |
| Map resolution | 2.09 |
| Fourier shell correlation threshold (Å) | 0.143 |
| B factor (Å²) | 59.2 |
| Resolution range (Å) | 2–3.5 |
| **Model refinement** | |
| Model composition | |
| Non-hydrogen atoms | 11,068 |
| Protein residues | 1455 |
| Ligands | 7 |
| Root mean square deviations | |
| Bond lengths (Å) | 0.005 |
| Bond angles (°) | 1.026 |
| Validation | |
| MolProbity score | 1.45 |
| Clashscore | 5.91 |
| Rotamer outliers (%) | 1.23 |
| CaBLAM outliers (%) | 1.46 |
| Rama-Z score | 0.63 |
| Ramachandran plot | |
| Favored (%) | 97.71 |
| Allowed (%) | 2.29 |
| Disallowed (%) | 0.00 |
| Model vs. data CC (mask) | 0.86 |

below at an average distance of 3.1 Å (s.d. 0.3 Å) consistent with its role in dehydrating the incoming K⁺ ion. Similarly, the S4 position is well coordinated from above at 3.1 Å (s.d. 0.2 Å) with only 2 oxygen ligands below, both of which come from side chain oxygen atoms instead of main-chain carbonyls. Additional densities provide coordination at both S1 and S4, which we have modeled as water given that their density levels are similar to the surrounding protein.

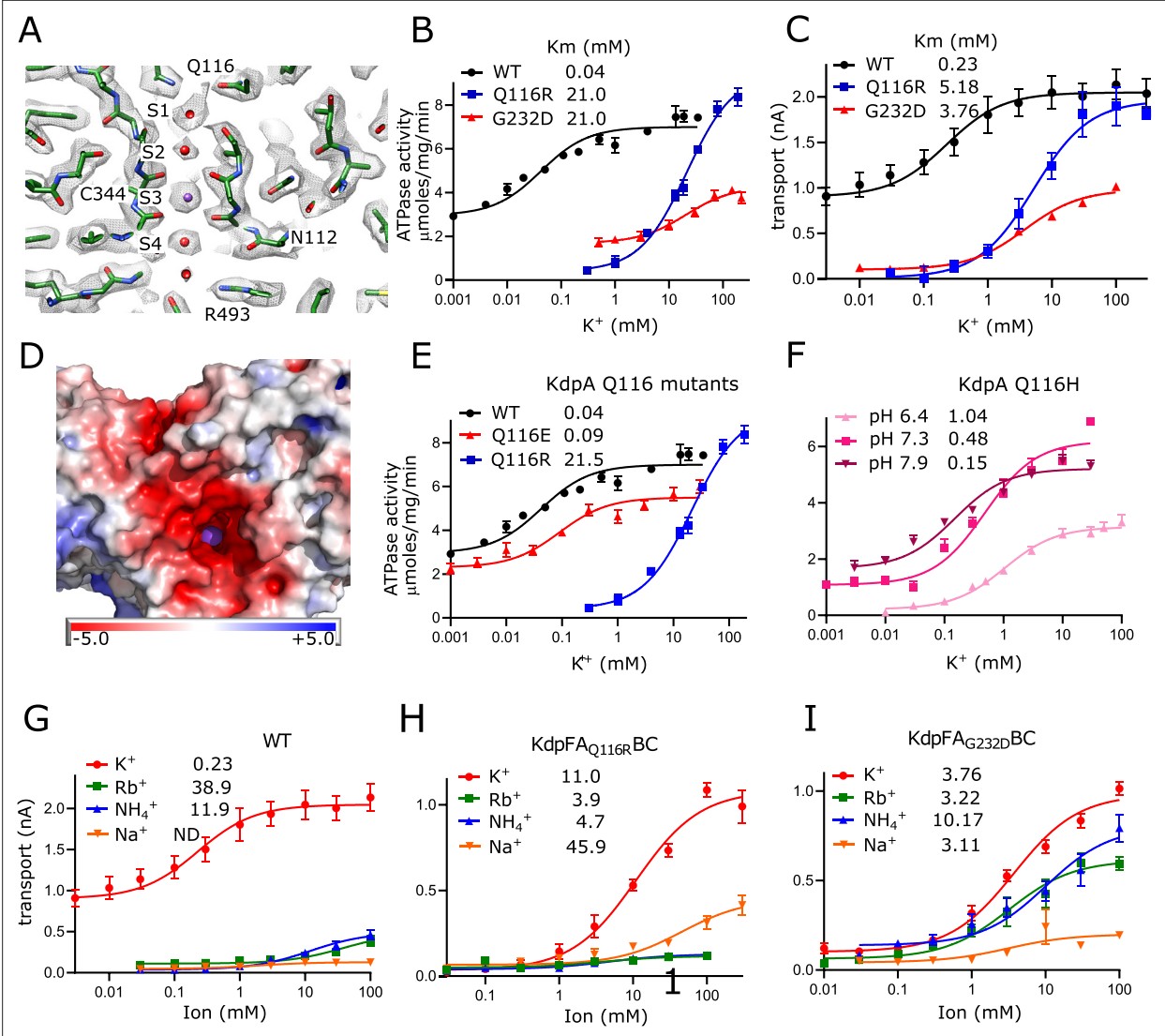

**Figure 2.** Selectivity filter of KdpA. (**A**) The map reveals densities at all sites (S1–S4) of the selectivity filter. Because the density at the S3 site is the highest in the entire map (*Figure 2—figure supplement 1A*), it has been assigned as a $K^+$ ion (purple sphere). Gray mesh corresponds to the sharpened map at 6.5 σ. (**B, C**) ATPase and transport assays (respectively) illustrate that Q116R and G232D mutations lower the apparent affinity of the pump (Km values in mM shown in the legend). Background activity is consistently observed for the WT, which is likely due to $K^+$ contamination in stock solutions (*Figure 1—figure supplement 1H*). The location of Gln116 is shown in panel (**A**), G232 is out of the plane in this image, but can be seen in *Figure 2—figure supplement 1B*. (**D**) Mapping of electrostatic charge indicates that the entrance to the selectivity filter of the WT pump is strongly negative, which would help attract positively charged ions. A $K^+$ ion is visible in the selectivity filter as a purple sphere. (**E, F**) Substitution of Glu116 has differing effects that reflect changes in net charge at the entrance to the selectivity filter. (**G–I**) Transport assays with various monovalent cations show that the strong selectivity of the WT pump is compromised by the G232D mutation but not by the Q116R mutation. *Table 2* shows measurements from transport assays that are broadly consistent with these ATPase data. Data are derived from 3–8 technical replicates as reflected in the source data and the error bars represent SEM. Raw data for the transport assays are shown in *Figure 2—figure supplements 2–4*.

The online version of this article includes the following source data and figure supplement(s) for figure 2:

**Source data 1.** Source data for *Figure 2B, C, E–I*, which includes all individual data points used for these plots.

**Figure supplement 1.** Selectivity filter of KdpA.

**Figure supplement 1—source data 1.** Source data for the histogram in *Figure 2—figure supplement 1A*.

**Figure supplement 2.** Current traces for WT KdpFABC.

**Figure supplement 2—source data 1.** Source data for *Figure 2—figure supplement 2* includes all the raw data from WT for the traces shown in this figure.

**Figure supplement 3.** Current traces for Q116R.

*Figure 2 continued on next page*

*Figure 2 continued*

**Figure supplement 3—source data 1.** Source data for *Figure 2—figure supplement 3* includes all the raw data from the Q116R mutant for the traces shown in this figure.

**Figure supplement 4.** Current traces for G232D.

**Figure supplement 4—source data 1.** Source data for *Figure 2—figure supplement 3* includes all the raw data from the G232D mutant for the traces shown in this figure.

The functional role of this selectivity filter has previously been probed by mutagenesis of Gln116 at the entrance and Gly232 in the middle. Specifically, Q116R and G232D substitutions were initially discovered by random mutagenesis during early characterization of the Kdp system (*Buurman et al., 1995*; *Epstein et al., 1978*) and have featured in many follow-up studies (*Dorus et al., 2001*; *Schrader et al., 2000*; *Silberberg et al., 2021*; *Sweet et al., 2020*; *van der Laan et al., 2002*). We have expanded these studies by comparing rates of ATP hydrolysis and K$^+$ transport to assess coupling, and by substituting a variety of amino acids to explore the role of electrostatics on ion binding. Consistent with previous work, K$^+$ titrations combined with measurement of ATPase (*Figure 2B*) and transport (*Figure 2C*) activities show that WT has affinity in the micromolar range which is severely reduced by mutations of either Gln116 or Gly232. Background activity in the absence of added K$^+$ was consistently observed for WT KdpFABC, which we attribute to K$^+$ contamination found in commercially available sources of ATP and MgCl$_2$ (*Figure 1—figure supplement 1h*). Nevertheless, the overall correspondence between ATPase and transport titrations indicates that energy coupling is maintained despite differences in apparent affinity. Structural work has shown that G232D is more disruptive to the structure of the selectivity filter (*Silberberg et al., 2021*), which may explain the lower V$_{max}$ seen for this mutant. Indeed, detergent-solubilized preparations proved to be relatively unstable during extended ATPase assay titrations at 25°C. Previous studies also showed a reduction of V$_{max}$ for Q116R, but that work employed the S162A mutation on KdpB to prevent post-translational phosphorylation of that residue and consequent formation of an inhibited complex (*Sweet et al., 2020*). Our use of the KdpD-knockout strain to produce unphosphorylated protein shows that Q116R can achieve WT levels of activity when the native Ser162 is retained.

Next, we tested the hypothesis that lower apparent affinity of the Q116R substitution was simply due to charge repulsion from an added positive charge at the entrance of the selectivity filter, whereas the G232D substitution results in a more severe disruption of the selectivity filter architecture. Indeed, mapping electrostatic charge on the surface of the WT protein shows an abundance of negative charge at the entrance to the selectivity filter, which would effectively attract substrate cations (*Figure 2D*). Consistent with such an electrostatic effect, we found that Q116E behaves comparably to WT, with substantial background activity and Km < 100 μM (*Figure 2E*). Furthermore, the Km for Q116H was pH dependent with decreased apparent affinity (higher Km) as the His became protonated and thus positively charged (*Figure 2F*). For G232D, we confirmed previous observations (*Schrader et al., 2000*; *van der Laan et al., 2002*) that this mutation not only reduced affinity, but also ion selectivity relative to WT and Q116R. Whereas the latter are high selectivity for K$^+$, G232D also displays robust activity with Rb$^+$ and NH$_4^+$ (*Figure 2G–I*). Notably, very little activity was seen with Na$^+$. In general, measurements of ATPase and transport were comparable, indicating that these mutants remain energetically coupled (*Table 2*).

## Vestibule in KdpA

Immediately after the selectivity filter there is a large cavity analogous to the vestibule seen in K$^+$ channels (*Doyle et al., 1998*). However, unlike channels where it leads directly to the cytosol, the vestibule of KdpFABC leads to the inter-subunit tunnel. There are many side chain oxygen atoms in this region that produce a generally hydrophilic environment, and our new structure reveals numerous discrete densities within this cavity (*Figure 3A*). Because their magnitude is comparable to or lower than the surrounding protein side chains, we model them as water molecules, thus producing a highly hydrated vestibule. Glu370 and Arg493 are the only potentially charged residues within the membrane core of KdpA and are positioned on either side of the S4 site. Glu370 lies at the entrance of the tunnel, whereas Arg493 lines the back wall of the vestibule, and we used mutagenesis in conjunction with ATPase and transport assays to probe their roles (*Figure 3C and D*, *Table 3*). Changes to Arg493 generally increase Km (lower apparent affinity) without affecting V$_{max}$, with Met substitution having

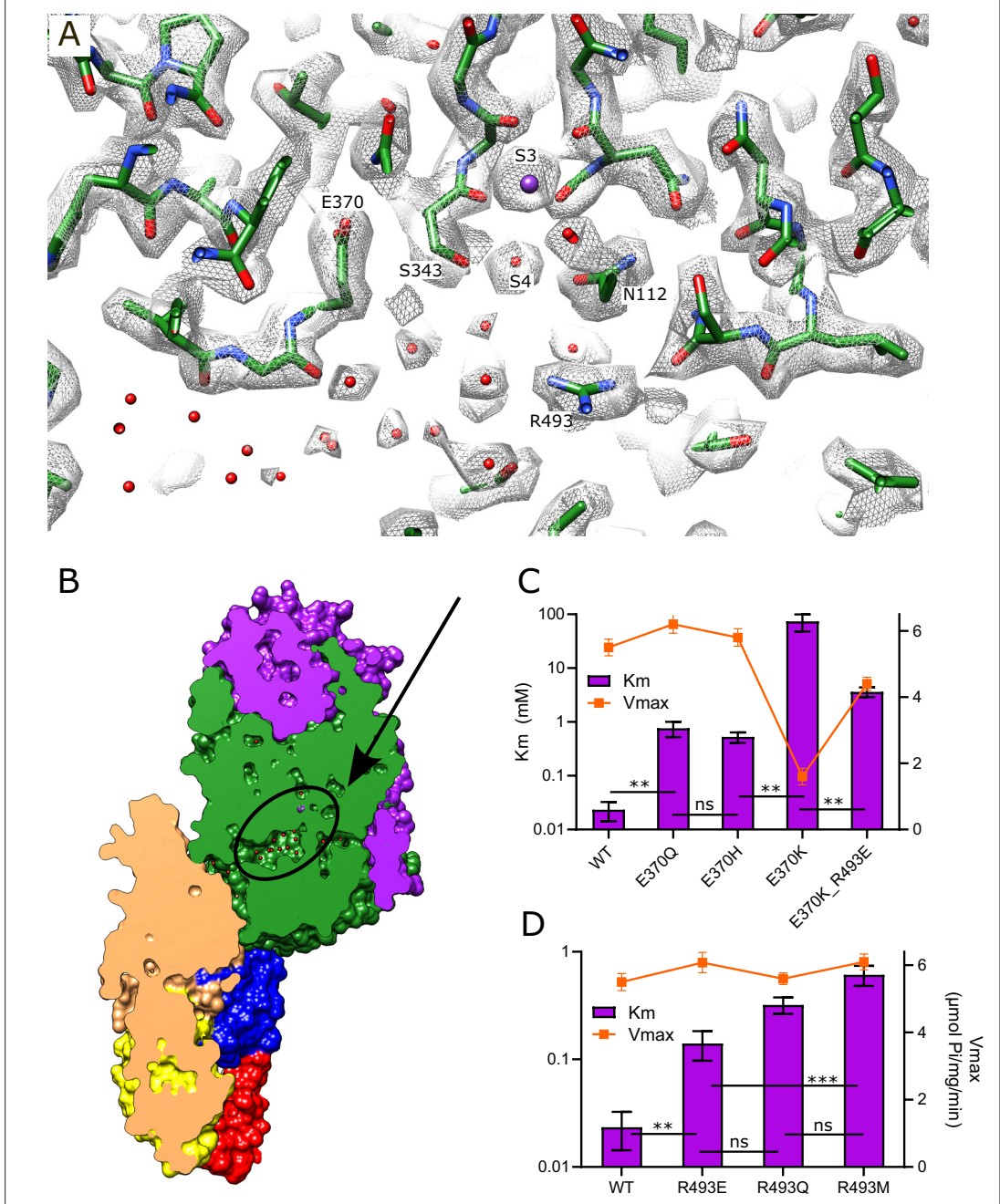

**Figure 3.** Vestibule of KdpA. (**A**) The map reveals numerous spherical densities immediately below the selectivity filter and leading into the tunnel that connects KdpA with KdpB. Glu370 and Arg493 are the only charged residues in the transmembrane domain of KdpA and reside on either side of this vestibule. Gray mesh corresponds to the sharpened map at 6.5 σ. (**B**) Slice through a surface rendering of the model for KdpFABC showing location of the vestibule pictured in panel (**A**). (**C**) ATPase assays of E370Q and E370H show that neutralization of charge on Glu370 reduces the apparent affinity without affecting $V_{max}$, but that reversing the charge with E370K almost completely abolishes activity. The double mutant E370K/R493E partially restores activity, but with much lower apparent affinity and $V_{max}$. (**D**) ATPase assays show that mutation of Arg493 has relatively modest effects on apparent affinity without affecting $V_{max}$. Note that the axis with Km is logarithmic. *Table 3* shows measurements from transport assays that are broadly consistent with these ATPase data. Data in panels (**C**) and (**D**) were derived from three technical replicates plotted with SEM; statistical significance was evaluated by one-way ANOVA analysis: **$p < 0.01$, ***$p < 0.001$ and ns indicates 'not significant'.

The online version of this article includes the following source data for figure 3:

**Source data 1.** Source data for *Figure 3C and D*, which includes all individual data points used for these plots.

**Table 2.** Summary of apparent affinities of the selectivity filter mutations measured by ATPase and transport assays.

| mutants | ATPase*<br>Km (mM) | transport†<br>Km (mM) |
|---|---|---|
| WT | | |
| $K^+$ | 0.04±0.01 | 0.23±0.10 |
| $Rb^+$ | 30.71±13.45 | 38.88±22.45 |
| $NH_4^+$ | - ‡ | - ‡ |
| $Na^+$ | - ‡ | - ‡ |
| G232D | | |
| $K^+$ | 21.17±2.66 | 3.76±0.58 |
| $Rb^+$ | - § | 3.22±1.06 |
| $NH_4^+$ | - § | 10.17±4.16 |
| $Na^+$ | - ‡ | - ‡ |
| Q116R | 21.45±2.66 | 5.19±0.12 |
| Q116E | 0.09±0.02 | 0.11±0.46 |

*Average of N=3 independent measurements with SEM.
†Average of N=4 independent measurements with SEM.
‡Little activity above baseline prevented Km determination.
§Protein instability in detergent solution interfered with accurate Km determination.

greater effect than charge reversal (R493E). Neutralization of Glu370 with a Gln or His substitution has a similar effect, but reversing the charge (E370K) significantly reduces both apparent affinity and $V_{max}$, suggesting an important functional role for negative charge in this region. The E370K/R493E double mutant partially restores $V_{max}$ but the apparent affinity is still compromised, suggesting that location of charge is essential for efficient movement of $K^+$ into the tunnel.

### Inter-subunit tunnel

A continuous tunnel has been documented in KdpFABC structures in the E1 conformation (*Huang et al., 2017*) and has been postulated to provide the conduction pathway for $K^+$ ions as they move from the selectivity filter in KdpA to the canonical ion binding sites in KdpB (*Stock et al., 2018*; *Sweet et al., 2021*). In our new structure, the discrete densities seen in the vestibule continue into this

**Table 3.** Summary of apparent affinities of the vestibule mutations measured by ATPase transport assays.

| mutants | ATPase*<br>Km (mM) | transport†<br>Km (mM) |
|---|---|---|
| WT | 0.023±0.01 | 0.23±0.10 |
| E370Q | 0.76±0.24 | 1.32±0.58 |
| E370H | 0.53±0.11 | 0.35±0.12 |
| E370K | 74.95±25.8 | - ‡ |
| E370K_R493E | 3.63±0.74 | 0.43±0.08 |
| R493E | 0.14±0.04 | 0.50±0.23 |
| R493Q | 0.32±0.06 | 1.52±0.68 |
| R493M | 0.61±0.13 | 0.47±0.08 |

*Average of N=3 independent measurements with SEM.
†Average of N=4 independent measurements with SEM.
‡Little activity above baseline prevents Km determination.

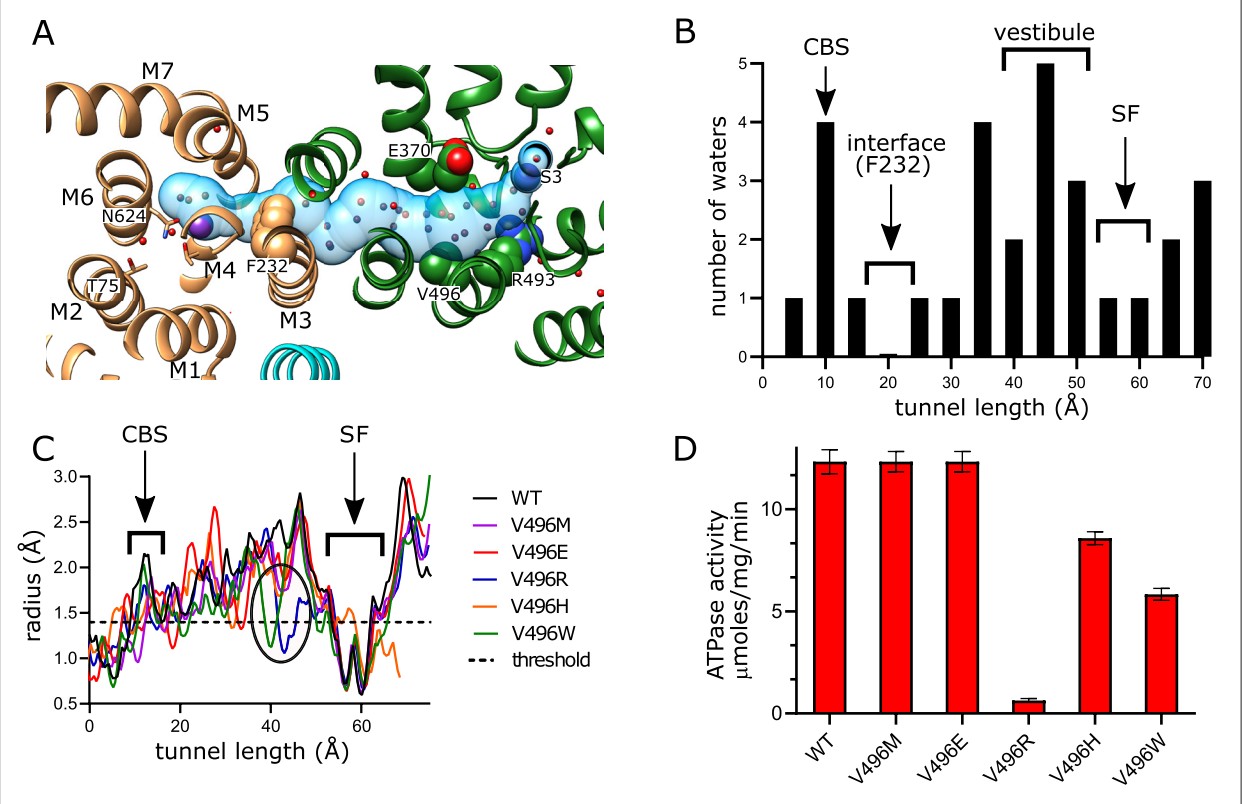

**Figure 4.** Tunnel connecting KdpA and KdpB. (**A**) The inter-subunit tunnel is shown as a transparent blue surface running from the selectivity filter on the right (S3) to the canonical binding site on the left (near Asn624). Non-protein densities within the tunnel have been modeled as water molecules (small spheres). Asp370, Arg493 in the vestibule and Phe232 at the subunit interface are shown as space-filling models and K+ as large purple spheres. As in other figures, KdpA helices are green and KdpB helices are brown; KdpF is cyan. (**B**) The distribution of water molecules within the tunnel shows that the vestibule is well populated, whereas the subunit interface is quite hydrophobic. Values along the x-axis correspond to the contour length along the tunnel. (**C**) Tunnel profiles for the experimentally determined model (WT) and predicted models for the Val496 mutants. The horizontal dotted line at a radius of 1.4 Å represents the conventional limit for passage of water or K+ ions. Note that both V496R and V496W constrict the tunnel. SF and CBS indicate locations of the selectivity filter and canonical binding site, respectively. (**D**) ATPase assays of the Val496 mutants in 150 mM K+ showing that introduction of positive charge (V496R) abolishes activity. Data was derived from six technical replicates and plotted with SEM. Raw data for the transport assays are shown in *Figure 4—figure supplement 1*.

The online version of this article includes the following source data and figure supplement(s) for figure 4:

**Source data 1.** Source data for *Figure 4B–D*, which includes individual data points used for these plots.

**Figure supplement 1.** Transport activity of Val496 mutants.

**Figure supplement 1—source data 1.** Source data for *Figure 4—figure supplement 1* includes all the data shown in this figure.

tunnel, forming a well-coordinated network with spacings between 2.5 and 3.5 Å. Whereas previous structures show 5–7 discrete densities extending from the vestibule into the tunnel (*Silberberg et al., 2022*; *Silberberg et al., 2021*; *Sweet et al., 2021*), our new structure reveals ~17 spherical densities in this region, all of comparable strength to the surrounding protein density. These densities, which we have modeled as water, are most prevalent near the vestibule, which is the wider part of the tunnel, but then disappear completely at the subunit interface near Phe232, which is the narrowest part of the tunnel and also distinctly hydrophobic (*Figure 4*).

In order to provide experimental evidence for K+ transport through the tunnel, we made a series of substitutions to Val496 in KdpA. This residue resides near the widest part of the tunnel and is fully exposed to its interior (*Figure 4A*). We made substitutions to increase its bulk (V496M and V496W) and to introduce charge (V496E, V496R, and V496H). We used the AlphaFold-3 artificial intelligence structure prediction program (*Jumper et al., 2021*) to generate structures of these mutants and to evaluate their potential impact on tunnel dimensions. This analysis predicts that V496W and V496R reduce the radius to well below the 1.4 Å threshold required for passage of K+ or water (*Figure 4C*);

V496E and V496M also constrict the tunnel, but to a lesser extent. Measurements of ATPase activity (*Figure 4D*) show that negative charge (V496E) or a longer side chain (V496M) has no apparent effect on ATPase activity. V496R, which introduces positive charge, almost completely abolishes activity. V496W and V496H reduce ATPase activity by about half, perhaps due to steric hindrance for the former and partial protonation for the latter. Transport activity of these mutants was also measured, but quantitative comparisons are hampered by potential inconsistency in reconstitution of proteoliposomes and in preparation of sensors for SSME. To account for differences in reconstitution, we compared ATPase activity and transport currents taken from the same batch of vesicles (*Figure 4— figure supplement 1A*). These data show that differences in ATPase activity of proteoliposomes were consistent with differences measured prior to reconstitution (*Figure 4D*). Transport activity, which was derived from multiple sensors, mirrored these ATPase activities, indicating that the Val496 mutants did not affect energy coupling, but simply modulated turnover rate of the pump.

## Canonical ion binding site in KdpB

KdpB belongs to the P-type ATPase superfamily where ion binding occurs in a membrane-embedded chamber next to a conserved proline along the M4 helix (Pro264 in KdpB). This proline interrupts the M4 helix and allows free carbonyls to form a binding site along its axis. For the well-studied P-type ATPases such as Sarcoendoplasmic Reticulum $Ca^{2+}$-ATPase (SERCA) and Sodium-Potassium ATPase (Na,K-ATPase), this canonical site is referred to as Ca2 and Na2, respectively, and additional ions are accommodated in the cavity between M4, M5, and M6 (Ca1 site for SERCA and Na1 plus Na3 sites for Na,K-ATPase) (*Kanai et al., 2013*; *Toyoshima et al., 2000*; *Figure 5—figure supplement 1*). In the original X-ray structure of an inhibited conformation of KdpFABC, a spherical density was present at the site corresponding to Ca2/Na2 and was modeled as water due to the lack of anomalous K signal at that location (*Huang et al., 2017*). In subsequent E1·ATP structures of KdpFABC stabilized with AMPPCP, densities were seen in the cavity harboring the Na1 site and were modeled either as water (*Sweet et al., 2021*) or $K^+$ (*Silberberg et al., 2021*). A recent structure captured under turnover conditions and ascribed to the E1~P conformation (7ZRK) showed a weak, elongated density leading toward the Ca2/Na2 site, which was modeled as $K^+$, but was not fully resolved from the end of Lys586 which resides in this region (*Silberberg et al., 2022*). A second E1~P structure (7ZRM) from the same study did not resolve any densities in this area. Our new structure unambiguously resolves a discrete spherical density precisely at the canonical Ca2/Na2 site that is well coordinated by main chain carbonyl atoms from M4 (Val260, Cys261, Ile263) as well as the side chains of Thr265 and Asn624 with an average distance of 2.8±0.2 Å (*Figure 5A*, *Figure 5—figure supplement 2A*). Given that this is the strongest density in KdpB, measuring 5.6× higher than the map densities shown in *Figure 5* (*Figure 5—figure supplement 2B*), we have modeled it as $K^+$. Additional densities within this chamber have similar magnitude to the surrounding protein residues and have been modeled as water, including two that coordinate the central $K^+$ at distances of 2.9 and 3.4 Å. These observations strongly support the canonical Ca2/Na2 site as the primary $K^+$ binding site for KdpB in the high energy E1~P·ADP conformation.

Asp583 and Lys586 are two conserved residues on M5 that have previously been shown to be essential for transport (*Bramkamp and Altendorf, 2005*). We made a series of substitutions to further probe their roles (*Figure 5B*). Most of these mutants show little to no activity, consistent with their crucial roles in shepherding $K^+$ through this binding site. This includes neutralization (D583N, K586Q) and reversal (D583K, K586E) of the charge. Swapping of the charge with a double mutant (D583K and K586D) also led to inactivation, showing that the geometry of the site is important. Consistent with earlier work on this mutant (*Bramkamp and Altendorf, 2005*), the D583A mutant retained some ATPase activity (30% of WT) but lacked transport, indicating that this particular mutation interfered with energy coupling.

## $K^+$ release site in KdpB

In other P-type ATPases, conformational changes convert the high-affinity canonical binding site to a low-affinity binding site with access to the opposite side of the membrane, thus allowing ions to be released directly from these sites (*Dyla et al., 2020*). For KdpFABC, however, analogous conformational changes have not been observed and the low-affinity release site for $K^+$ ions is not well established. We previously proposed a novel mechanism (*Sweet et al., 2021*) in which Lys586 swings

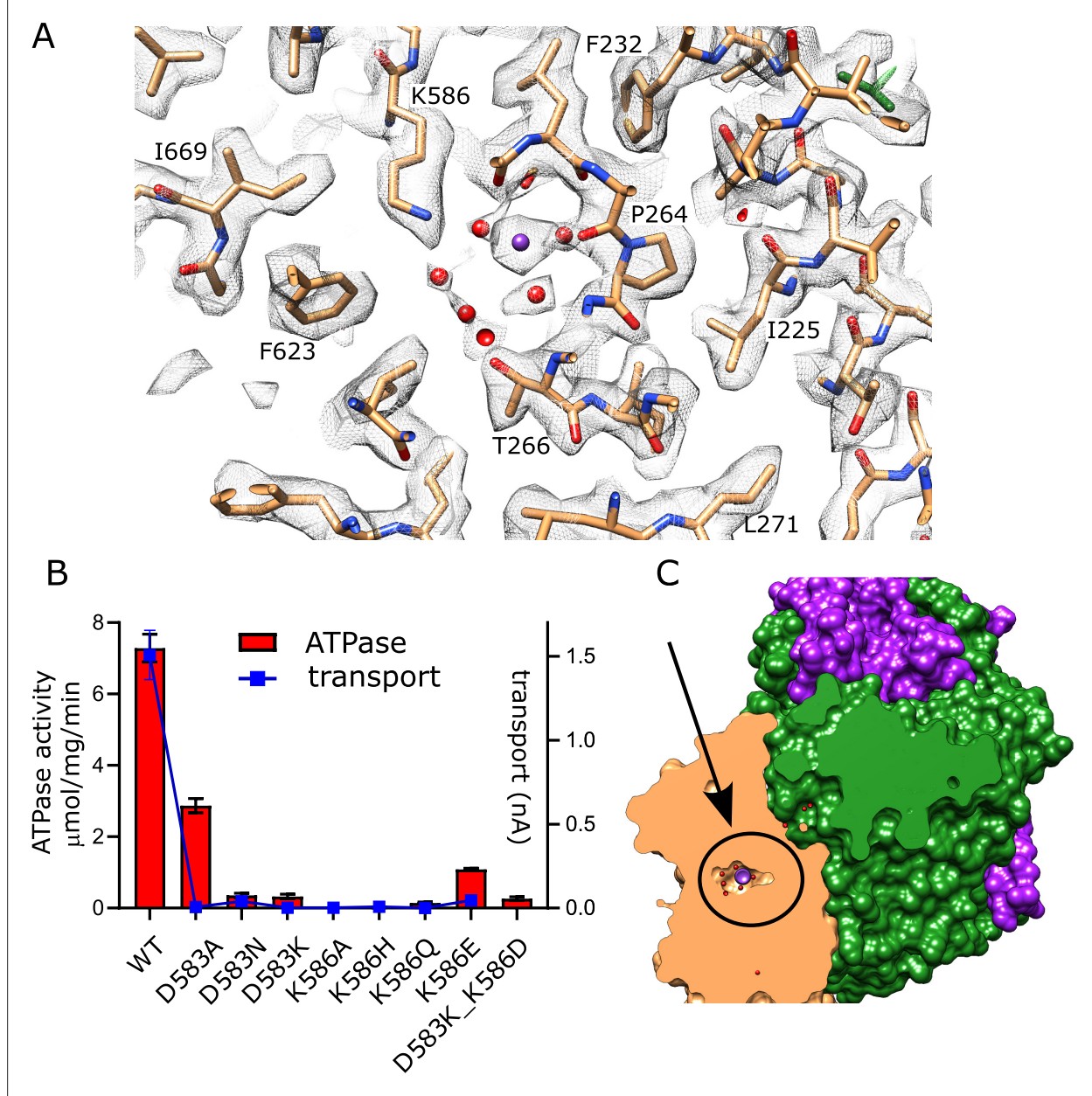

**Figure 5.** Canonical binding site in KdpB. (**A**) Several non-protein densities are visible at the canonical binding site, the most prominent of which occupies the conserved Na2/Ca2 site from Na,K-ATPase and SERCA (*Figure 5—figure supplement 1*). This site has the highest density in KdpB (*Figure 5—figure supplement 2B*) and thus has been modeled as $K^+$ (purple sphere), whereas the others are modeled as water (red spheres). Gray mesh corresponds to the sharpened map at 5.8 σ. (**B**) Asp583 and Lys586 are the only charged residues in the transmembrane domain of KdpB. ATPase and transport assays in 150 mM $K^+$ show that mutation of these residues is generally not tolerated, except for the D583A mutation that generates an uncoupled phenotype (ATPase activity without transport). Data was derived from 3 to 8 technical replicates as reflected in the source data and error bars represent SEM. Raw data for the transport assays are shown in *Figure 5—figure supplement 3*. (**C**) Slice through a surface rendering of the model for KdpFABC showing location of the CBS pictured in panel (**A**).

The online version of this article includes the following source data and figure supplement(s) for figure 5:

**Source data 1.** Source data for *Figure 5B*, which includes all individual data points used for this plot.

**Figure supplement 1.** Canonical binding sites of P-type ATPases.

**Figure supplement 2.** Properties of the canonical binding site in KdpB.

**Figure supplement 2—source data 1.** Source data for *Figure 5—figure supplement 1* includes all the data shown in panels B and C.

**Figure supplement 3.** Current traces from CBS mutants.

**Figure supplement 3—source data 1.** Source data for *Figure 5—figure supplement 3* includes all the raw data traces shown in this figure.

into the canonical Ca2/Na2 binding site (previously called the Bx site) in the transition to an E2 state, thus pushing K⁺ into an alternate chamber (previously called the B2 site) between M1, M2, and M4 (*Figure 6—figure supplement 1A*). This alternate chamber has access to the cytoplasm via a water-filled cavity, and the relatively weak coordination of ions within this chamber is consistent with a low-affinity release site. Although our new structure does not represent the state relevant for release, a non-protein density is seen in this chamber next to Thr75, which was previously postulated as a potential ligand for ions or water in this site (*Figure 6A*).

To experimentally test this proposed release site, we substituted Thr75 with either Asp or Lys and measured effects on both ATPase and transport activity (*Figure 6C–E*). For these measurements, we varied the pH to alter the charge at this site and looked for corresponding effects on the activity. At neutral pH, T75D produced K⁺-dependent ATPase and transport activities comparable to WT, indicating that the Asp substitution does not interfere with turnover or energy coupling. However, maximal activity of T75D is shifted toward more acidic pHs and, unlike WT, falls off dramatically at basic pH. Assuming that the pKa of T75D is shifted by this intramembrane environment, these results are consistent with the idea that negative charge generated by deprotonation of the Asp residue interferes with K⁺ release. Indeed, pKa shifts have previously been studied both experimentally (*Gayen et al., 2016*; *Isom et al., 2010*; *Morrison et al., 2015*) and computationally (*Henderson et al., 2020*; *Panahi and Brooks, 2015*), showing pKa values up to 9 for buried acidic residues. In the case of T75D, we applied PROPKA (*Li et al., 2005*) on a predicted model for this mutant, which suggested a substantial shift from the expected pKa of aspartate in solution (~3.6) toward neutral pH in this intramembrane environment (pKa of 6.8).

As expected, the T75K mutation behaves quite differently. At pH<8, T75K has robust ATPase activity that does not depend on the presence of K⁺ and does not generate any transport current (*Figure 6E*). This behavior is consistent with a loss of energy coupling. At pH>8, K⁺-dependence of ATPase activity is restored and a small transport current is measured, suggesting partial recovery as charge on the Lys is reduced. Although the pKa of the lysine side chain in solution is 10.8, it would be expected to shift toward neutral pH in the membrane environment, again supported by a PROPKA calculation based on the predicted model for T75K (pKa of 6.3). Inspection of this predicted model reveals that the side chain of the lysine is inserted into the canonical binding site, analogous to the behavior of Lys586 in the E2 conformation (*Figure 6—figure supplement 1B*). This suggests that T75K acts as a 'built-in cation' (*Pedersen et al., 2007*) that is detected at the canonical binding site to constitutively induce ATPase activity by KdpB, even in the absence of K⁺. Leu72 is another residue deeper in this release pocket and the L72D substitution was also tested. Although overall activity is diminished, it displays a pH dependence similar to T75D (*Figure 6—figure supplement 1C*).

## Discussion

We have used structure determination, mutagenesis, and activity measurements to study the transport pathway for K⁺ ions through KdpFABC. For cryo-EM analysis, we purified WT protein free from inhibitory effects of post-translational serine phosphorylation, reconstituted it into lipid nanodiscs, and imaged this sample under turnover conditions, thus providing the most native conditions visualized by structural biology to date. The largest population of particles adopted the E1~P·ADP conformation, consistent with it preceding the rate-limiting step in the transport cycle. The resolution of the corresponding structure (2.1 Å) is much improved over previous structural studies, revealing new features relevant to this pathway.

The inter-subunit tunnel is arguably one of the most intriguing elements of the KdpFABC complex. Although it has been postulated to conduct K⁺, direct experimental evidence has been hard to come by. However, our mutagenesis studies now firmly support this mechanism and illustrate key properties of this K⁺ conduit. To start, the external surface of the selectivity filter carries negative charge, thus serving to attract ions. Mutations that reduce the net charge lower the apparent affinity, thus explaining the well-known effects of the Q116R mutant (*Buurman et al., 1995*). Once in the selectivity filter, the ion appears to have a strong preference for the S3 site. This conclusion is supported by X-ray anomalous scattering (*Huang et al., 2017*; *Madapally et al., 2025*), the consistent presence of density at this site in previous lower-resolution structures (*Silberberg et al., 2022*; *Silberberg et al., 2021*; *Sweet et al., 2021*), and the exceptionally strong density in current map (*Figure 2—figure supplement 1*). Furthermore, coordination geometries in the selectivity filter are consistent with preferential

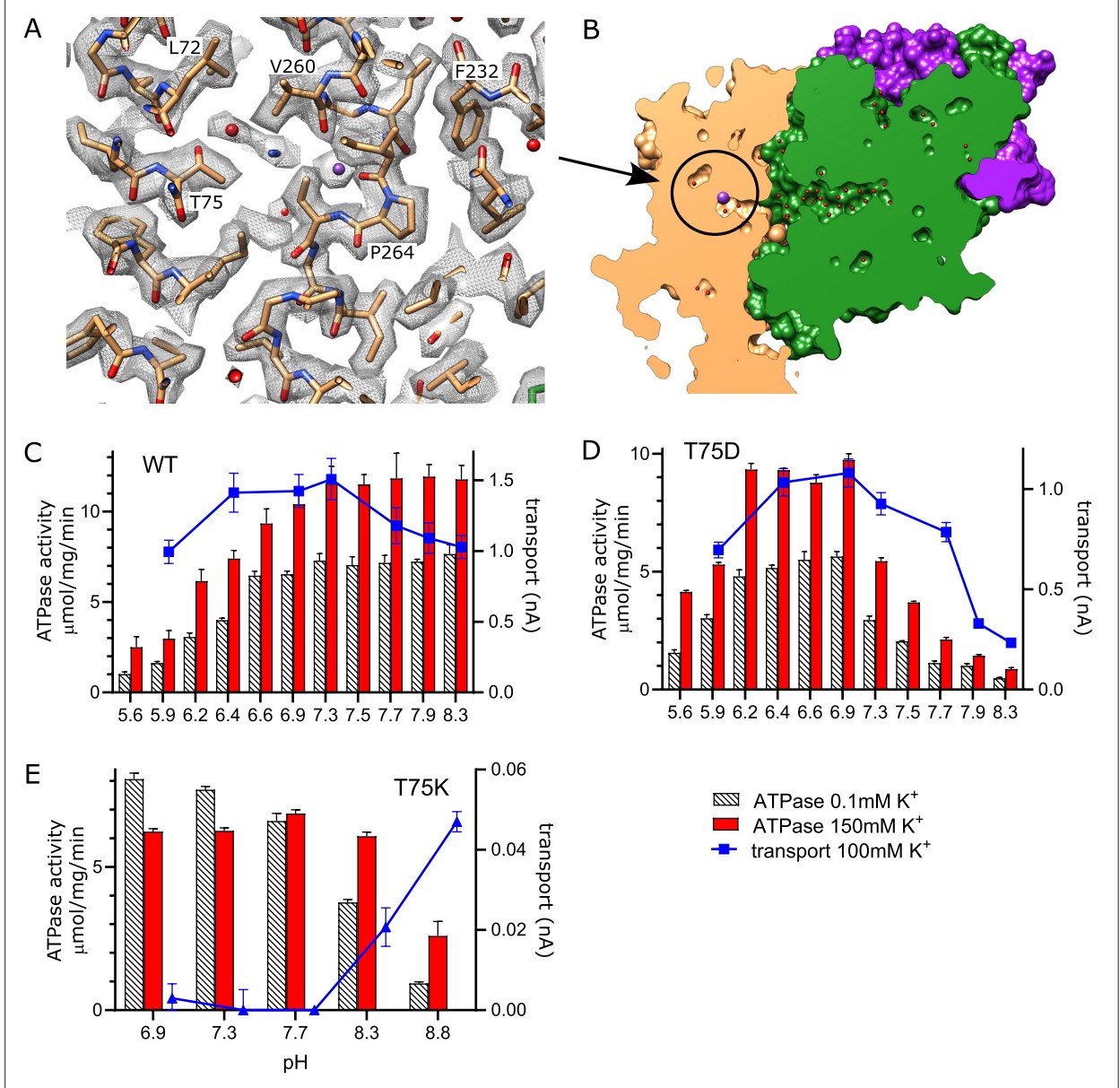

**Figure 6.** Exit site from KdpB. (**A**) The map shows a water-filled cavity next to Thr75 that leads to the cytoplasm (sharpened map at 6.5 σ). (**B**) Slice through a surface rendering of the model for KdpFABC showing location of the exit site pictured in panel (**A**). (**C, D**) pH dependence of the T75D mutant is shifted relative to WT, consistent with the idea that high pH generates negative charge in this cavity and inhibits K⁺ release. (**E**) The T75K substitution not only interferes with transport, but also produces an uncoupled phenotype in which ATPase activity is not dependent on K⁺. At high pH, there is a small recovery of transport activity as well as K⁺-dependence of ATPase activity, suggesting that these effects result from the positive charge from the lysine substitution. Data was derived from six technical replicates and plotted with SEM. Raw data for the transport assays are shown in *Figure 6—figure supplement 2*.

The online version of this article includes the following source data and figure supplement(s) for figure 6:

**Source data 1.** Source data for *Figure 6C–E*, which includes all individual data points used for these plots.

**Figure supplement 1.** Exit site for K⁺ from KdpB.

**Figure supplement 1—source data 1.** Source data for *Figure 6—figure supplement 1* includes all the raw data shown in panel C.

**Figure supplement 2.** Current traces from exit site mutants.

**Figure supplement 2—source data 1.** Source data for *Figure 6—figure supplement 2* includes all the raw data traces shown in this figure.

binding at S3 and suggest that high apparent affinity and selectivity toward K$^+$ comes from this site, and not from the canonical binding site in KdpB. Unlike K$^+$ channels, the relatively slow turnover of KdpFABC (10–100 s$^{-1}$) allows for equilibrium binding at the selectivity filter, which may explain its ability to distinguish K$^+$ from its congener Rb$^+$ (*Huang et al., 2017*; *Liu and Lockless, 2013*). As in previous work (*Schrader et al., 2000*; *van der Laan et al., 2002*), the G232D mutation resulted in loss of selectivity and lowered affinity. Given that Gly232 resides in the middle of the selectivity filter, these results are readily understood by its disruptive effects on the S3 site (*Silberberg et al., 2021*). Our observation that G232D is capable of coupled transport for NH$_4^+$ and Rb$^+$ confirms not only that the selectivity filter governs ion selection, but that the pump subunit, KdpB, is relatively promiscuous.

Like K$^+$ channels, the selectivity filter opens into a water-filled vestibule. Unlike channels, the vestibule in KdpFABC does not lead to the cytoplasm, but instead forms the entrance to the inter-subunit tunnel. Nevertheless, mutations to Glu370 and Arg493 indicate that, like channels, this region influences the conductance of ions (*Naranjo et al., 2016*). These are the only charged residues in the membrane core of KdpA and, although they do not physically interact and do not project directly into the vestibule, they likely create an electrostatic field in the low dielectric environment of the membrane. Arg493, which is more highly conserved, is at the back of the vestibule, whereas Glu370 is along the direction of ion travel. Neutralization of either residue or reversal of their charge increases Km, though V$_{max}$ is unaffected. Indeed, alignment of diverse KdpA sequences shows that Glu370 is neutralized (Asn or Gln) in ~50% of the organisms (*Huang et al., 2017*). Interestingly, reversal of charge on the fully conserved Arg493 (R493E) has relatively modest effects, but reversal of Glu370 (E370K) essentially abolishes transport, perhaps indicating a role for an electrostatic field in moving K$^+$ through this region.

Non-protein spherical densities have been seen within the vestibule and tunnel in previous maps, but the current map reveals many more, likely due to its higher resolution. The identity of these densities has been contentious, sometimes modeled as a string of 7–9 K$^+$ ions (*Silberberg et al., 2022*; *Silberberg et al., 2021*) whereas our previous structural models feature water molecules in similar positions (*Sweet et al., 2021*). Despite difficulty in distinguishing atomic species by cryo-EM, we continue to favor water molecules within this confined space for several reasons: first, a structure of the G232D mutant with K$^+$ substituted for Rb$^+$ did not change the normalized scattering density within the tunnel, as was seen at the S3 and canonical binding sites (*Silberberg et al., 2021*). Second, unconstrained all-atom MD simulations concluded that the tunnel accommodates at most one K$^+$ ion at a time, presumably due to electrostatic repulsion (*Madapally et al., 2025*). These simulations, which documented spontaneous entry of water into the vestibule and tunnel during the equilibration phase, resulted in a distribution of waters within the tunnel that is broadly consistent with our new high-resolution map. Third, an anomalous X-ray signal for K$^+$ has been observed at the S3 site but not in this region (*Huang et al., 2017*; *Madapally et al., 2025*). Fourth, based on the volume of the cavity in KdpA, a single K$^+$ ion would correspond to a concentration of 3.4 M, suggesting that multiple ions would exceed the solubility limit especially in the absence of counterions. Finally, map densities within the tunnel were either of comparable strength or weaker than surrounding side chain atoms, unlike at S3 and canonical binding sites. Although it is possible that weaker density could represent low occupancy K$^+$ ions, we favor a mechanism whereby individual K$^+$ ions occupy the tunnel transiently as they transit between the selectivity filter and the canonical binding site. However, the energetics of this process remain obscure as the tunnel becomes narrow and dewetted as it approaches the KdpA/KdpB subunit interface. Furthermore, calculations of free energy indicate a massive energy barrier (~22 kcal/mol) at this interface, suggesting that we are missing a key conformation that would facilitate passage (*Madapally et al., 2025*).

Both previous (*Silberberg et al., 2022*) and current cryo-EM studies indicate that the E1~P state is the prevalent conformational state under turnover conditions. This observation suggests that the ensuing step, E1~P to E2-P, is rate limiting, as also seen in other P-type ion pumps (*Kühlbrandt, 2004*). Our higher resolution map shows an exceedingly strong, well-resolved density at the canonical binding site along the axis of the unwound M4 helix, providing compelling evidence that, like other P-type pumps, the transport ion is bound at this canonical site in this state. Several additional, weaker densities are present in the adjacent pocket between M4, M5, and M6, coordinated either by this bound K$^+$ or by the surrounding residues. Their relative strength is consistent with water molecules, indicating that the stoichiometry of transport is likely to be 1 K$^+$ per ATP. The presence of these waters

in the relatively large pocket together with the innate flexibility of M4 is consistent with promiscuity of this site. Indeed, when the selectivity filter is disrupted by the G232D mutation, the complex appears to be perfectly capable of transporting a broader range of cations.

Similar to the Glu370/Arg493 charge pair in KdpA, Asp583 and Lys586 are the only charged residues in the membrane core of KdpB. Although they are not seen to interact directly in our structure, they coordinate accessory waters associated with the canonical binding site. Previous molecular dynamics simulations (*Silberberg et al., 2021*) indicate that Asp583 couples with Phe232 to form a 'proximal binding site' for K$^+$ ions. Based on these simulations, these authors proposed a mechanism whereby neutralization of this site either by ion binding or by D583A substitution served to stimulate ATPase activity. Indeed, earlier work on D583A (*Bramkamp and Altendorf, 2005*) as well as current data demonstrate uncoupling, in which K$^+$-independent ATPase activity was observed even though transport was abolished. A plausible explanation for this stimulation is seen in the behavior of Lys586 in previous structures of the E2·Pi state (7BGY and 7BH2) (*Sweet et al., 2021*). In these structures, M5 undergoes a conformational change that pushes the side chain of Lys586 into the CBS. As a consequence of the D583A mutation, this Lys could be freed to act as a built-in counter ion as in related P-type ATPases ZntA (*Wang et al., 2014*) and AHA2 (*Pedersen et al., 2007*). In regard to the proximal binding site and the partnering "distal binding site" on the KdpA-side of the subunit interface, our structure does not show densities at either site and thus does not provide any support for the related mechanism. In any case, in the WT complex, it seems likely that Asp583 exerts allosteric control over Lys586 and ensures that its movement into the binding site is coordinated with the transition from E1~P to E2·Pi, thus leading to displacement of K$^+$ from the CBS and release to the cytoplasm.

In P-type ATPases such as SERCA and Na,K-ATPase, ions are bound and released from the canonical binding pocket next to the proline of M4. The ATPase cycle is associated with changing affinity and accessibility of this site to alternate sides of the membrane. Although KdpB belongs to this superfamily, it operates in a substantially different way with ions entering the canonical binding site from the inter-subunit tunnel rather than directly from the cytosol or periplasm. Stock et al. initially proposed that release occurred directly from the canonical site via a pathway between M4, M5, and M6 (*Stock et al., 2018*). However, this proposed pathway is very narrow and does not depend on the enzymatic state of the protein. We previously proposed an alternative exit site for K$^+$ between M1, M2, and M4 that aligns with pathways seen in other P-type ATPases (*Dyla et al., 2020*). In KdpB, this alternative pathway appears to have low affinity and is constitutively open to the cytosol. We proposed that the key step in transport was moving the ion from the canonical site into this exit site, from which it would spontaneously exit (*Sweet et al., 2021*). Current results from Thr75 mutants provide the first experimental evidence supporting this hypothesis. In particular, the pH dependence of T75D indicates that introduction of negative charge into this exit site interferes with transport, most likely by preventing release of the ion from this site. This conclusion relies on the assumption that the pKa of the Asp is shifted toward neutral pH due to the membrane environment, which is supported by theoretical estimation of pKa. On the other hand, T75K generated an uncoupled phenotype with WT levels of ATPase activity even in the absence of K$^+$. Structure prediction for this mutant suggests that the substituted lysine inserts into the canonical binding site, similar to the insertion of Lys586 seen in the E2·Pi conformation (*Figure 6—figure supplement 1*). This hypothesis extends the idea of lysines acting as built-in ions capable of stimulating futile ATPase cycles. More generally, this behavior is consistent with KdpB acting as a rather non-specific cation-stimulated ATPase capable of recognizing its natural ligand, K$^+$, as well as Rb$^+$, NH$^+$ , or the positive charge from nearby lysine residues.

## Summary

*Figure 7* summarizes features of the transport pathway elucidated by the current work. Effects of mutants indicate that a negative surface charge attracts K$^+$ into the selectivity filter of KdpA where it binds at the S3 site, which is the primary determinant of ion specificity. Like K$^+$ channels, the selectivity filter opens into a vestibule that is filled with water, where ions are attracted into the inter-subunit tunnel by Glu370. The proximal end of this tunnel is filled with water but becomes hydrophobic as it approaches the interface with KdpB, where ions are dewetted and encounter a large energy barrier. The current structure represents E1~P·ADP in which the K$^+$ has arrived at the canonical binding site in KdpB, next to the conserved Pro264. In the ensuing, rate-limiting step to E2·Pi, the conserved Lys586 on M5 moves into the canonical binding site, thus displacing the ion into a novel exit site next

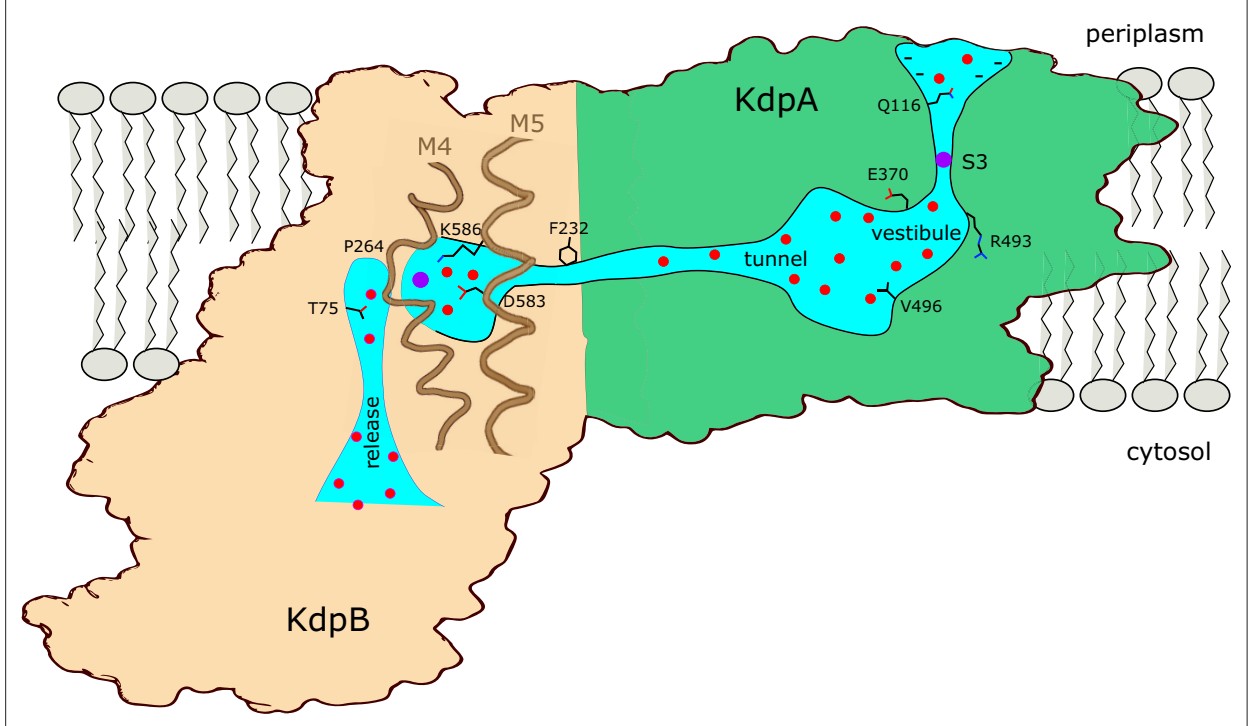

**Figure 7.** Conduction pathway of KdpFABC. Diagram illustrating features of the conduction pathway seen in the E1~P·ADP map together with residues that have been mutated in this study. Ions (purple spheres) are attracted from the periplasm to the mouth of the selectivity filter of KdpA by negative surface potential. Selectivity and affinity of the pump is governed by tight coordination of the ion at the S3 site. Ions become rehydrated upon release into a vestibule filled with water (red spheres) next to charged residues Glu370 and Arg493. Val496 resides in the widest part of the tunnel where introduction of positive charge abolishes transport. Phe232 resides at the subunit interface where there is a hydrophobic barrier that is essential to ion occlusion. K[+] binding to the canonical binding site next to Pro264 on M4 in KdpB triggers phosphorylation of Asp307 followed by allosteric movements of M5 that drive Lys586 to displace the ion into the exit site next to Thr75. A general lack of ligands in this exit site is consistent with low affinity, thus promoting K[+] release to the cytoplasm against the electrochemical gradient.

to Thr75. This scenario differs from other P-type pumps which release ions directly from their initial binding site. Nevertheless, T75 as well as the entire cytoplasmic portion of the M2 helix are well conserved in Kdp systems (*Huang et al., 2017*), consistent with a crucial role in the transport pathway.

## Materials and methods
### Expression and purification of KdpFABC mutants
Expression and purification of KdpFABC has been described previously (*Hussein et al., 2024*; *Sweet et al., 2021*). Briefly, we used *E. coli* strain TK2281 (thi, rha, lacZ, nagA, trkA405, trkD1, Δ(kdp-FABCDE)81) to express WT and the various mutant proteins from a pBAD plasmid. This strain lacks KdpD, which has recently been shown to phosphorylate Ser162 on KdpB and thus inhibit its activity under conditions used for overexpression of mutant proteins (*Silberberg et al., 2024*; *Sweet et al., 2020*). For expression, cells were cultured in Luria Broth (LB) media to $OD_{600}$ of 0.6 and then expression of KdpFABC was induced with 1.3 mM L-arabinose for 3 h. After cell lysis, the membrane fraction was solubilized by overnight incubation in 50 mM Tris pH 7.5, 600 mM NaCl, 10 mM MgCl₂, 10% glycerol, 1 mM tris (2-carboxyethyl) phosphine (TCEP), and 1.2% n-decyl-β-maltoside (DM) at 4°C (20 ml of buffer for each gram of the membrane pellet). The sample was then loaded onto a Ni-NTA HiTrap affinity column (Cytiva, Marlborough, MA) equilibrated with 50 mM Tris pH 7.5, 600 mM NaCl, 10 mM MgCl₂, 10% glycerol, 1 mM TCEP, and 0.15% DM and eluted with an imidazole gradient. Peak fractions were pooled, concentrated, and further purified with a size-exclusion column (Superdex 200 increase 10/300 GL, GE Healthcare) using a running buffer containing 100 mM NaCl, 25 mM Tris pH 7.5, 10% glycerol, 1 mM TCEP, and 0.15% DM (*Figure 1—figure supplement 1*).

## Expression and purification of MSP protein

A circularized membrane scaffold protein designated spNW25 was used for nanodisc formation (**Zhang et al., 2021**). For expression, *E. coli*, BL21 STAR (DE3) cells carrying spNW25 on a pET vector were cultured in LB media at 37°C, 200 rpm to an $OD_{600}$ of 0.7 followed by addition of 0.25 mM isopropyl β-D-1-thiogalactopyranoside for 16 h at 16°C. Cells were collected by centrifugation at 3500 × $g$ for 20 min at 4°C and then resuspended in 50 mM Tris-HCl pH 8, 100 mM NaCl, 5% glycerol, and 2 mM β-mercaptoethanol supplemented with protease inhibitor cocktail tablet. Cells were lysed by three passages through an Emulsiflex C3 high-pressure homogenizer (Avestin, Ottawa Canada). Lysed cells were centrifuged at 12,000 × $g$ for 45 min and the supernatant was loaded onto a 5 ml Ni-NTA column equilibrated with 50 mM Tris-HCl (pH 8), 400 mM NaCl, 5% glycerol, 20 mM imidazole. spNW25 was eluted after washing the column with buffers containing 25 mM and 50 mM imidazole followed by a 50–500 mM imidazole gradient. Fractions containing purified protein were pooled, dialyzed against 50 mM Tris-HCl pH 8, 100 mM NaCl, 5% glycerol, 2 mM β-mercaptoethanol, then aliquoted, frozen, and stored at –80°C.

## Preparation of lipid nanodisc

Stock solutions of lipid were prepared by mixing 1-palmitoyl-2-oleoyl-glycero-3-phosphocholine (POPC) and 1,2-dioleoyl-sn-glycero-3-phosphate (DOPA), Avanti Polar Lipids, Alabaster, AL in 20:1 weight ratio in chloroform. A thin film was produced by drying this mixture under Argon gas for 1–2 h and this film was then resuspended in 20 mM Tris pH 7.4, 100 mM KCl, 0.5 mM TCEP, and 10 mg/ml Triton X-100 at a final lipid concentration of 5 mg/ml. After incubation at room temperature for 30 min, 0.18 mg of lipid was mixed with 0.48 mg of KdpFABC and 0.72 mg of spNW25 (total volume of 0.5 ml) and incubated for 60 min. Detergent was removed by sequential addition of BioBeads (BioRad Hercules, CA): 1 mg for 30 min, 1 mg for 30 min, and 40 mg overnight. This preparation was optimized by evaluating size distribution of nanodiscs by HPLC size-exclusion chromatography and negative stain electron microscopy. In addition, the composition of nanodiscs was determined by SEC-MALS analysis (**Figure 1—figure supplement 1**).

For preparation of cryo-EM samples, reconstituted nanodiscs were filtered with a 0.22 μm syringe filter and loaded onto a Superdex 200 size exclusion column that was equilibrated with 20 mM Tris pH 7.4, 100 mM KCl, and 0.5 mM TCEP. Peak fractions were concentrated to 4.25 mg/ml using a 50 kDa cutoff centrifugal concentrator (AMICON, Millipore Sigma, Burlington, MA). The concentrated sample was aliquoted into PCR tubes and mixed with MgATP buffer to yield a final concentration of 2 mM $MgCl_2$, 1 mM ATP, and 3.4 mg/ml KdpFABC nanodisc. 3 μl of sample was added to glow-discharged EM grids (C-Flat 1.2/1.3-4Cu-50; Protochips, Inc) and blotted under 100% humidity at 4°C prior to plunging into liquid ethane using a Vitrobot (Thermo Fisher Scientific Inc, Bridgewater, NJ).

## Activity assays

ATPase rates were determined using the coupled enzyme assay (**Warren et al., 1974**). The standard buffer was composed of 75 mM TES-Tris pH 7.3, 7.5 mM $MgCl_2$, 0.5 mM phosphoenolpyruvate, 2.4 mM ATP, 0.8 mM NADH, 0.3% DM, 9.6 U/ml lactate dehydrogenase, and 9.6 U/ml pyruvate kinase. For potassium titration, 25 mM TES-Tris, 2.5 mM $MgCl_2$, and 1 mM ATP were used. All reagents were acquired from Sigma-Aldrich (St. Louis, MO).

For ion transport, KdpFABC was reconstituted into proteoliposomes composed of a 4:1 weight ratio of POPC and DOPA as previously described (**Hussein et al., 2024**). Briefly, liposomes were prepared by resuspending a thin film of lipid in 50 mM HEPES-Tris pH 7.1, 5 mM $MgSO_4$, followed by three freeze–thaw cycles and extrusion using a 0.4 nm polycarbonate filter (Whatman-Cytiva). Triton X-100 was added to liposomes at lipid to detergent weight ratio of 2.5:1, followed by addition of KdpFABC to yield a final lipid concentration of 5 mg/ml and a lipid-to-protein ratio of 5:1. The detergent was slowly removed by the sequential addition BioBeads.

Ion transport was measured using solid-supported membrane electrophysiology as implemented by the SURFE²RN1 (Nanion Technologies, Livingston, NJ). Proteoliposomes were diluted 1:10 with inactivation buffer (50 mM HEPES-Tris, pH 7.1, 5 mM $MgSO_4$, 150 mM $K_2SO_4$) and bath sonicated four times for 10 s each. Gold sensors were passivated with 1,2-diphytanoyl-sn-glycero-3-phosphocholine according to the manufacturer's instructions. After adding 50 μl of inactivation buffer to each sensor, 10 μl of diluted proteoliposomes were added and sensors were centrifuged at 2500 × $g$ for 30 min at

4°C. For measurement of transport, an automated protocol was used in which the sensor was washed with inactivation buffer for 1 s, followed by 1 s of activation buffer (inactivation buffer with 300 µM of ATP) and again with inactivation buffer for 1 s. Several approaches were taken to ensure the stability of sensors over time. First, we performed repeated measurements under conditions reflecting $V_{max}$ and, after ~14 replicates, generally observed <10% loss in signal. Second, although data was taken from titrations progressing from the highest ion concentrations toward the lowest, we routinely also performed titrations in the reverse direction: from lowest to highest ion concentrations. Generally speaking, the maximal signal from the two protocols was comparable, though data from the former tended to be more reproducible. Finally, for ions that produced negligible transport, K$^+$ was introduced at the end of the titration to assess activity of the sample. In general, data reflects a minimum of six transport measurements from two different sensors. The raw data obtained from the SURFE$^2$RN1 and used for quantification of transport are shown in several figure supplements: *Figure 2—figure supplements 2–4*, *Figure 5—figure supplement 3*, and *Figure 6—figure supplement 2*.

Some experiments involved a fixed ion concentration, whereas others required titration. For the former, that is, mutations in the tunnel (Val496), CBS (Asp583 and Lys586) and exit site (Thr75), sensors were prepared with 100 mM K$^+$ (i.e., 50 mM K$_2$SO$_4$). For ion titrations, 80 µM of nigericin was mixed with the diluted proteoliposomes prior to addition to sensors. Nigericin is an electroneutral ionophore that facilitates equilibration of monovalent cations across the membrane (*Prabhananda and Ugrankar, 1991*). Sensors were then prepared in the absence of added K$^+$, and a rinsing step followed by a 4 min incubation period was included to the protocol to change the ion concentration. For pH titration, buffers containing 50 mM K$_2$SO$_4$ were prepared at different pHs; for pH <6.4, MES-Tris buffer was used instead of HEPES-Tris. Sensors were initially prepared at pH 7.3, and pH was changed by including a rinsing step followed by a 3 min incubation. At the end of the titration, transport at the initial pH was measured again to ensure that there was no loss of signal.

It is well established that current measurements by SSME can reflect the steady-state process of transport as well as the pre-steady-state process of ion binding within the dielectric field of the membrane (*Bazzone et al., 2023*; *Bazzone et al., 2022*). These two signals can be distinguished by assessing the decay time of the signal, or the full width of the peak at half maximum. SSME signals from P-type pumps are typically dominated by sustained transport currents that display decreasing decay time constants, or decreased peak width, as the substrate concentration is increased (*Hussein et al., 2024*; *Tadini-Buoninsegni et al., 2006*). However, we did observe more transient binding currents for some of the mutants of KdpFABC. For example, the selectivity filter mutant Q116R produced a robust binding current but no observable transport current for Rb$^+$ and NH$_4^+$ (*Figure 2—figure supplement 3*). In order to circumvent this problem, we chose to quantify transport by measuring the current at 1.25 s, at which point the binding current had dissipated (*Bazzone et al., 2023*). All data was plotted and analyzed using GraphPad Prism10 software (GraphPad Software Inc, San Diego CA).

Potassium analysis of stock solutions was conducted using a simultaneous inductively coupled plasma-mass spectrometry (si-ICP-MS), known as Mattauch–Herzog-ICP-MS (MH-ICP-MS) instrument (SPECTRO MS, SPECTRO Analytical Instruments GmbH, Kleve, Germany) (*Rabieh et al., 2020*). Samples were acidified to 2% (vol/vol) with 65% Suprapur nitric acid (analytical-reagent grade, Merck, Darmstadt, Germany) and transferred via a Cetac autosampler (ASX-560, Teledyne Cetac Technologies, Omaha, NE, USA) to the nebulizer of MH-ICP-MS instrument. Each sample was analyzed in triplicate. A single-element potassium standard solution (Inorganic Ventures, Christiansburg, VA, USA) was used to prepare five K standard solutions ranging from blank to 2500 parts per billion, which were used for constructing the calibration curve. This standard was also used to prepare control and spiked samples, which were analyzed at the beginning and end of the run. The average potassium recoveries for the control and spiked (pyruvate kinase enzyme) samples were 95.5% and 91.2%, respectively.

## Cryo-EM structure determination

Samples of WT KdpFABC in lipid nanodiscs were imaged on a Krios G3i electron microscope equipped with Selectris imaging filter and Falcon4i detector (Thermo Fisher Scientific, Inc) at a pixel size of 0.93 Å and total dose of ~50 electrons/Å². A total of ~50,000 images were collected from four grids prepared from a single sample, which were processed with cryoSPARC v4.6 (*Punjani et al., 2017*). After removing images with crystalline ice, excessive contamination, or imaging artifacts, the remaining images were divided into groups of 2000–6000 for preliminary processing (*Figure 1—figure*

*supplement 2*). Particles were picked from each group using TOPAZ (*Bepler et al., 2019*), extracted with a box size of 335 Å with threefold binning (120 pixels), and subjected to two rounds of 2D classification to remove false positives. Selected particles were subjected to successive rounds of ab initio reconstruction with C1 symmetry and three classes, with the best two classes advancing to the next round. Particles selected from ab initio jobs were reextracted without binning (box size of 360 pixels) and used for successive rounds of heterogeneous refinement and 3D classification to separate the various conformations adopted under these active turnover conditions. Non-uniform refinement was ultimately used to produce structures from homogeneous classes. The largest class, corresponding to the E1~P·ADP conformation is being reported on here.

For structure refinement, we started with coordinates for KdpA-Q116R/KdpB-S162A mutant bound to AMPPCP (PDB 7LC3). After reversing the mutations, replacing AMPPCP with ADP and adding phosphate to Asp307, the model was fitted as a rigid body to the map using CHIMERA (*Pettersen et al., 2004*). This model was adjusted to the experimental map using COOT (*Emsley et al., 2010*) and then subjected to multiple rounds of real space refinement in PHENIX (*Adams et al., 2010*). The A- and N-domain of KdpB displayed somewhat lower resolution (*Figure 1—figure supplement 2*), presumably due to their flexibility. As a result, these domains were refined against the unsharpened map, which displayed better connectivity in the peripheral structural elements. These two cytoplasmic domains were then joined to the membrane domains in COOT to produce the final model. Model and map depictions were done in CHIMERA (*Pettersen et al., 2004*), CHIMERAX (*Meng et al., 2023*), and PyMOL Molecular Graphics System (Schrödinger, LLC).

For analysis of the water tunnel, we used Caver Analyst 2.0 Beta (*Jurcik et al., 2018*). Tunnel profiles were derived from a starting point near KdpB-Ile628 using the following settings: approximation 12, minimum probe radius 0.5, clustering threshold 3.5, shell depth 2.5, shell radius 4.5, residues included AA and 9Y0. Python scripts were used to identify water molecules within the tunnel and to calculate its volume (https://doi.org/10.5281/zenodo.16928540). To evaluate structural effects of Val496 mutants, we used AlphaFold 3 (*Jumper et al., 2021*) to predict structures and then analyzed the tunnel profile with the above parameters. To estimate the pKa values of the exit site mutants, we generated structures of the mutants using AlphaFold 3, which were then uploaded to the online version of PropKa (*Li et al., 2005*).

## Acknowledgements

We thank Dr. Sasan Rabieh from the College of Dentistry at New York University for analysis of $K^+$ contamination by inductively coupled plasma mass spectrometry. Screening of cryo-EM samples was performed at NYU Langone Health's Cryo-Electron Microscopy Laboratory (RRID:SCR_019202), which is partially supported by the Laura and Isaac Perlmutter Cancer Center Support Grant NIH/NCI P30CA016087 and NIH Grant R01NS108151. A portion of this research was supported by NIH grant R24GM154185 and performed at the Pacific Northwest Center for Cryo-EM (PNCC) with assistance from Sean Mulligan. This project received funding from Novo Nordisk Foundation grant NNF24OC0088380 and from the European Research Council (ERC) under the European Union's Horizon 2020 research and innovation program (grant agreement no. 101000936) to BPP and from the National Institutes of Health (grant R35GM144109) to DLS.

## Additional information

### Funding

| Funder | Grant reference number | Author |
| --- | --- | --- |
| National Institute of General Medical Sciences | R35GM144109 | David L Stokes |
| Novo Nordisk Fonden | NNF24OC0088380 | Bjørn P Pedersen |
| European Research Council | 101000936 | Bjørn P Pedersen |
| National Cancer Institute | P30CA016087 | David L Stokes |

| Funder | Grant reference number | Author |
|--------|------------------------|--------|
| National Institute of Neurological Disorders and Stroke | R01NS108151 | David L Stokes |
| National Institute of General Medical Sciences | R24GM154185 | David L Stokes |

The funders had no role in study design, data collection and interpretation, or the decision to submit the work for publication.

### Author contributions

Adel Hussein, Data curation, Formal analysis, Investigation, Visualization, Methodology, Writing – original draft, Writing – review and editing; Xihui Zhang, Investigation, Methodology; Bjørn P Pedersen, Conceptualization, Validation, Investigation, Writing – review and editing; David L Stokes, Conceptualization, Data curation, Software, Formal analysis, Supervision, Funding acquisition, Validation, Visualization, Methodology, Writing – original draft, Project administration, Writing – review and editing

### Author ORCIDs

David L Stokes (iD) https://orcid.org/0000-0001-5455-8163

Reviewer #3 (Public review): https://doi.org/10.7554/eLife.107397.4.sa1
Author response https://doi.org/10.7554/eLife.107397.4.sa2

## Additional files

### Supplementary files

MDAR checklist

### Data availability

The cryo-EM maps have been deposited in the Electron Microscopy Data Bank under accession number EMD-70308 and the corresponding atomic model has been deposited in the Protein Data Bank with accession number 9OC4. Python scripts used to identify water molecules within the tunnel and to calculate its volume have been deposited in GitHub and archived at Zenodo with the following DOI: https://doi.org/10.5281/zenodo.16928540. All data generated or analyzed during this study are included in the manuscript and supporting files; source data files have been provided for all relevant figures.

The following datasets were generated:

| Author(s) | Year | Dataset title | Dataset URL | Database and Identifier |
|-----------|------|---------------|-------------|-------------------------|
| Hussein AK, Zhang X, Pedersen BP, Stokes DL | 2025 | High-resolution cryo-EM structure of KdpFABC in the E1P-ADP state in lipid nanodisc | https://doi.org/10.2210/pdb9OC4/pdb | Worldwide Protein Data Bank, 10.2210/pdb9OC4/pdb |
| Hussein A, Zhang X, Pedersen BP, Stokes DL | 2025 | High-resolution cryo-EM structure of KdpFABC in the E1P-ADP state in lipid nanodisc | https://www.ebi.ac.uk/emdb/EMD-70308 | Electron Microscopy Data Bank, EMD-70308 |
| Stokes DL | 2025 | Python programs to analyze protein cavities generated by CAVER | http://doi.org/10.5281/zenodo.16928540 | Zenodo, 10.5281/zenodo.16928540 |

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

# Appendix 1

## Appendix 1—key resources table

| Reagent type (species) or resource | Designation | Source or reference | Identifiers | Additional information |
|---|---|---|---|---|
| Gene (*Escherichia coli*) | *KdpFABC* | *Huang et al., 2017* | *Uniprot IDs: kdpA: P03959 kdpC: P03961 kdpB: P03960 kdpF: P36937* | |
| Strain, strain background (*Escherichia coli*) | TK2281 | Wolfgang Epstein | Protein expression strain: thi, rha, lacZ, nagA, trkA405, trkD1, Δ(kdpFABCDE)81 | 10.1016/0076-6879(88)57113-6 |
| Strain, strain background (*Escherichia coli*) | *BL21 STAR (DE3)* | Invitrogen, Carlsbad, CA | Protein expression strain | |
| Recombinant DNA reagent | pBAD | Invitrogen, Carlsbad, CA | Empty vector (plasmid) | |
| Recombinant DNA reagent | pET28a-spNW25 | Addgene ID: 173484 | Plasmid for expression of nanodisc scaffolding protein | *Zhang et al., 2021* |
| Sequence-based reagent | G232D_F | Primer was ordered from Integrated DNA Technologies (IDT) | PCR primer | GCTCGGTACTAACGACGGTGGCTTCTTTAAT |
| Sequence-based reagent | G232D_R | Primer was ordered from Integrated DNA Technologies (IDT) | PCR primer | ATTAAAGAAGCCACCGTCGTTAGTACCGAGC |
| Sequence-based reagent | Q116E_For | Primer was ordered from Integrated DNA Technologies (IDT) | PCR primer | CACCAATACCAACTGGGAATCTTATAGCGGTGAAACCACGTTG |
| Sequence-based reagent | Q116E_R | Primer was ordered from Integrated DNA Technologies (IDT) | PCR primer | CTATAAGATTCCCAGTTGGTATTGGTGACAAAGCTGAC |
| Sequence-based reagent | Q116H_F | Primer was ordered from Integrated DNA Technologies (IDT) | PCR primer | CAATACCAACTGGCACTCTTATAGCGGTGAAACCACGTTG |
| Sequence-based reagent | Q116H_R | Primer was ordered from Integrated DNA Technologies (IDT) | PCR primer | GCTATAAGAGTGCCAGTTGGTATTGGTGACAAAG |
| Sequence-based reagent | R493M_F | Primer was ordered from Integrated DNA Technologies (IDT) | PCR primer | GTTTGTCGGTATGTTCGGGGTGATTATCCCGG |
| Sequence-based reagent | R493M_R | Primer was ordered from Integrated DNA Technologies (IDT) | PCR primer | CCCGAACATACCGACAAACATGCAGAAC |
| Sequence-based reagent | R493E_F | Primer was ordered from Integrated DNA Technologies (IDT) | PCR primer | GTTTGTCGGTGAGTTCGGGGTGATTATCCCGGTG |
| Sequence-based reagent | R493E_R | Primer was ordered from Integrated DNA Technologies (IDT) | PCR primer | CCCGAACTCACCGACAAACATGCAGAACGC |
| Sequence-based reagent | R493Q_F | Primer was ordered from Integrated DNA Technologies (IDT) | PCR primer | GTTTGTCGGTCAGTTCGGGGTGATTATCCCGGTG |
| Sequence-based reagent | R493Q_R | Primer was ordered from Integrated DNA Technologies (IDT) | PCR primer | CCCGAACTGACCGACAAACATGCAGAACG |
| Sequence-based reagent | E370Q_F | Primer was ordered from Integrated DNA Technologies (IDT) | PCR primer | CAAATTGGTCAAGTGGTGTTCGGCGG |

*Appendix 1 Continued on next page*

*Appendix 1 Continued*

| Reagent type (species) or resource | Designation | Source or reference | Identifiers | Additional information |
| --- | --- | --- | --- | --- |
| Sequence-based reagent | E370Q_R | Primer was ordered from Integrated DNA Technologies (IDT) | PCR primer | GAACACCACTTGACCAATTTGCATCAGCCACATCAGCCACATCG |
| Sequence-based reagent | E370K_F | Primer was ordered from Integrated DNA Technologies (IDT) | PCR primer | CAAATTGGTAAAGTGGTGTTCGGCGG |
| Sequence-based reagent | E370K_R | Primer was ordered from Integrated DNA Technologies (IDT) | PCR primer | CGAACACCACTTTACCAATTTGCATCAGCCAC |
| Sequence-based reagent | E370H_F | Primer was ordered from Integrated DNA Technologies (IDT) | PCR primer | CAAATTGGTCACGTGGTGTTCGGCGGTGTC |
| Sequence-based reagent | E370H_R | Primer was ordered from Integrated DNA Technologies (IDT) | PCR primer | GAACACCACGTGACCAATTTGCATCAGCCACATCG |
| Sequence-based reagent | V496M_F | Primer was ordered from Integrated DNA Technologies (IDT) | PCR primer | GCTTCGGGATGATTATCCCGGTGATGGCAATTG |
| Sequence-based reagent | V496M_R | Primer was ordered from Integrated DNA Technologies (IDT) | PCR primer | GGGATAATCATCCCGAAGCGACCGACAAAC |
| Sequence-based reagent | V496E_F | Primer was ordered from Integrated DNA Technologies (IDT) | PCR primer | GCTTCGGGGAGATTATCCCGGTGATGGCAATTG |
| Sequence-based reagent | V496E_R | Primer was ordered from Integrated DNA Technologies (IDT) | PCR primer | GGGATAATCTCCCCGAAGCGACCGACAAAC |
| Sequence-based reagent | V496R_F | Primer was ordered from Integrated DNA Technologies (IDT) | PCR primer | GCTTCGGGCGGATTATCCCGGTGATGGCAATTG |
| Sequence-based reagent | V496R_R | Primer was ordered from Integrated DNA Technologies (IDT) | PCR primer | GGGATAATCCGCCCGAAGCGACCGACAAAC |
| Sequence-based reagent | V496H_F | Primer was ordered from Integrated DNA Technologies (IDT) | PCR primer | GCTTCGGGCACATTATCCCGGTGATGGCAATTGC |
| Sequence-based reagent | V496H_R | Primer was ordered from Integrated DNA Technologies (IDT) | PCR primer | GGGATAATGTGCCCGAAGCGACCGACAAAC |
| Sequence-based reagent | V496W_F | Primer was ordered from Integrated DNA Technologies (IDT) | PCR primer | CTTCGGGTGGATTATCCCGGTGATGGC |
| Sequence-based reagent | V496W_R | Primer was ordered from Integrated DNA Technologies (IDT) | PCR primer | GGGATAATCCACCCGAAGCGACCGA |
| Sequence-based reagent | D583A_F | Primer was ordered from Integrated DNA Technologies (IDT) | PCR primer | CAGCATTGCCAACGCTGTGGCGAAATACTTCG |
| Sequence-based reagent | D583A_R | Primer was ordered from Integrated DNA Technologies (IDT) | PCR primer | CGAAGTATTTCGCCACAGCGTTGGCAATGCTG |
| Sequence-based reagent | D583N_F | Primer was ordered from Integrated DNA Technologies (IDT) | PCR primer | CATTGCCAACAATGTGGCGAAATACTTCGCCAT |

*Appendix 1 Continued on next page*

*Appendix 1 Continued*

| Reagent type (species) or resource | Designation | Source or reference | Identifiers | Additional information |
|---|---|---|---|---|
| Sequence-based reagent | D583N_R | Primer was ordered from Integrated DNA Technologies (IDT) | PCR primer | CGCCACATTGTTGGCAATGCTGAAGGTGGTC |
| Sequence-based reagent | D583K_F | Primer was ordered from Integrated DNA Technologies (IDT) | PCR primer | CATTGCCAACAAGGTGGCGAAATACTTCGCCATTATTC |
| Sequence-based reagent | D583K_R | Primer was ordered from Integrated DNA Technologies (IDT) | PCR primer | CGCCACCTTGTTGGCAATGCTGAAGGTGGTCAG |
| Sequence-based reagent | K586A_F | Primer was ordered from Integrated DNA Technologies (IDT) | PCR primer | CCAACGATGTGGCGGCATACTTCGCCATTATTC |
| Sequence-based reagent | K586A_R | Primer was ordered from Integrated DNA Technologies (IDT) | PCR primer | CCAACGATGTGGCGGCATACTTCGCCATTATTC |
| Sequence-based reagent | K586H_F | Primer was ordered from Integrated DNA Technologies (IDT) | PCR primer | GATGTGGCGCATTACTTCGCCATTATTCCGGCG |
| Sequence-based reagent | K586H-R | Primer was ordered from Integrated DNA Technologies (IDT) | PCR primer | GATGTGGCGCATTACTTCGCCATTATTCCGGCG |
| Sequence-based reagent | K586Q_F | Primer was ordered from Integrated DNA Technologies (IDT) | PCR primer | GATGTGGCGCAATACTTCGCCATTATTCCGGCG |
| Sequence-based reagent | K586Q_R | Primer was ordered from Integrated DNA Technologies (IDT) | PCR primer | GCGAAGTATTGCGCCACATCGTTGGCAATG |
| Sequence-based reagent | K586E_F | Primer was ordered from Integrated DNA Technologies (IDT) | PCR primer | GATGTGGCGGAATACTTCGCCATTATTCCGGCG |
| Sequence-based reagent | K586E_R | Primer was ordered from Integrated DNA Technologies (IDT) | PCR primer | GCGAAGTATTCCGCCACATCGTTGGCAATG |
| Sequence-based reagent | T75D_F | Primer was ordered from Integrated DNA Technologies (IDT) | PCR primer | GTGGATCGACGTACTGTTCGCTAATTTCGC |
| Sequence-based reagent | T75D_R | Primer was ordered from Integrated DNA Technologies (IDT) | PCR primer | GAACAGTACGTCGATCCACAGCCAACCGC |
| Sequence-based reagent | T75K_F | Primer was ordered from Integrated DNA Technologies (IDT) | PCR primer | GTGGATCAAGGTACTGTTCGCTAATTTCGCCG |
| Sequence-based reagent | T75K_R | Primer was ordered from Integrated DNA Technologies (IDT) | PCR primer | GCGAACAGTACCTTGATCCACAGCCAACCGCT |
| Sequence-based reagent | L72D_F | Primer was ordered from Integrated DNA Technologies (IDT) | PCR primer | CGGTTGGGACTGGATCACCGTACTGTTCG |
| Sequence-based reagent | L72D_R | Primer was ordered from Integrated DNA Technologies (IDT) | PCR primer | GTGATCCAGTCCCAACCGCTAATGGCC |
| Peptide, recombinant protein | spNW25 MSP protein | This study. Protein was expressed and purified as described in the 'Materials and methods' section | Nanodisc scaffolding protein | *Zhang et al., 2021* |

*Appendix 1 Continued on next page*

*Appendix 1 Continued*

| Reagent type (species) or resource | Designation | Source or reference | Identifiers | Additional information |
|---|---|---|---|---|
| Chemical compound, drug | 1-palmitoyl-2-oleoyl-glycero-3-phosphocholine (POPC) | Avanti Polar Lipids, Alabaster, AL | Lipid | |
| Chemical compound, drug | 1,2-dioleoyl-sn-glycero-3-phosphate (DOPA) | Avanti Polar Lipids, Alabaster, AL | Lipid | |
| Chemical compound, drug | 1,2-diphytanoyl-sn-glycero-3-phosphocholine | Avanti Polar Lipids, Alabaster, AL | Lipid | |
| Chemical compound, drug | n-Decyl-β-D-Maltopyranoside | anatrace | Detergent | |
| Software, algorithm | cryoSPARC | Structura Biotechnology | RRID:SCR_016501 | |
| Software, algorithm | RELION | *Scheres, 2012* | 5.0 | https://www3.mrc-lmb.cam.ac.uk/relion/index.php/Main_Page |
| Software, algorithm | Chimera | *Pettersen et al., 2004* | RRID:SCR_004097 | |
| Software, algorithm | ChimeraX | *Meng et al., 2023* | RRID:SCR_015872 | |
| Software, algorithm | PyMOL | Schrodinger | RRID: SCR_000305 | |
| Software, algorithm | PHENIX | *Adams et al., 2010* | RRID:SCR_014224 | |
| Software, algorithm | COOT | *Emsley et al., 2010* | RRID:SCR_014222 | |
| Software, algorithm | PRISM | GraphPad | RRID:SCR_002798 | |
| Software, algorithm | CAVER | https://caver.cz/ | Caver Analyst 2.0 | |
| Software, algorithm | cavervolume.py | This study | Python script | 10.5281/zenodo.16928540 |
| Software, algorithm | water_profile.py | This study | Python script | 10.5281/zenodo.16928540 |
| Software, algorithm | ASTRA 5.3.4.20 | Wyatt Technology | MALS analysis software | |
| Others | SURFE2R N1 sensors 3 mm | Nanion Technologies, Livingston, NJ | SSME measurement instrument | |
| Others | C-Flat 1.2/1.3-4Cu-50 grids | Protochips, Inc | Cryo-EM grids | |

