## [Editor Report · eLife Assessment]

This article revisits the well-studied KdpFABC potassium transport system from bacteria with a **convincing** set of new higher resolution structures, a protein expression strategy that permits purification of the active wildtype protein, and insight obtained from mutagenesis and activity assays. The thorough and thoughtful mechanistic analyses make this a **valuable** contribution to the membrane transport field.

---

## [Referee Report · Reviewer #3 (Public review)]

Summary:

By expressing protein in a strain that is unable to phosphorylate KdpFABC, the authors achieve structures of the active wildtype protein, capturing a new intermediate state, in which the terminal phosphoryl group of ATP has been transferred to a nearby Asp, and ADP remains covalently bound. The manuscript examines the coupling of potassium transport and ATP hydrolysis by a comprehensive set of mutants. The most interesting proposal revolves around the proposed binding site for K+ as it exits the channel near T75. Nearby mutations to charged residues cause interesting phenotypes, such as constitutive uncoupled ATPase activity, leading to a model in which lysine residues can occupy/compete with K+ for binding sites along the transport pathway.

Strengths:

The high resolution (2.1 Å) of the current structure is impressive, and allows many new densities in the potassium transport pathway to be resolved. The authors are judicious about assigning these as potassium ions or water molecules, and explain their structural interpretations clearly. In addition to the nice structural work, the mechanistic work is thorough. A series of thoughtful experiments involving ATP hydrolysis/transport coupling under various pH and potassium concentrations bolsters the structural interpretations and lends convincing support to the mechanistic proposal. The SSME experiments are rigorous.

---

## [Author Response]

The following is the authors’ response to the previous reviews

**Reviewer #2 (Public review):**
Summary:The paper describes the high-resolution structure of KdpFABC, a bacterial pump regulating intracellular potassium concentrations. The pump consists of a subunit with an overall structure similar to that of a canonical potassium channel and a subunit with a structure similar to a canonical ATP-driven ion pump. The ions enter through the channel subunit and then traverse the subunit interface via a long channel that lies parallel to the membrane to enter the pump, followed by their release into the cytoplasm.The work builds on the previous structural and mechanistic studies from the authors' and other labs. While the overall architecture and mechanism have already been established, a detailed understanding was lacking. The study provides a 2.1 Å resolution structure of the E1-P state of the transport cycle, which precedes the transition to the E2 state, assumed to be the ratelimiting step. It clearly shows a single K+ ion in the selectivity filter of the channel and in the canonical ion binding site in the pump, resolving how ions bind to these key regions of the transporter. It also resolves the details of water molecules filling the tunnel that connects the subunits, suggesting that K+ ions move through the tunnel transiently without occupying welldefined binding sites. The authors further propose how the ions are released into the cytoplasm in the E2 state. The authors support the structural findings through mutagenesis and measurements of ATPase activity and ion transport by surface-supported membrane (SSM) electrophysiology.
**Reviewer #3 (Public review):**
Summary:By expressing protein in a strain that is unable to phosphorylate KdpFABC, the authors achieve structures of the active wildtype protein, capturing a new intermediate state, in which the terminal phosphoryl group of ATP has been transferred to a nearby Asp, and ADP remains covalently bound. The manuscript examines the coupling of potassium transport and ATP hydrolysis by a comprehensive set of mutants. The most interesting proposal revolves around the proposed binding site for K+ as it exits the channel near T75. Nearby mutations to charged residues cause interesting phenotypes, such as constitutive uncoupled ATPase activity, leading to a model in which lysine residues can occupy/compete with K+ for binding sites along the transport pathway.Strengths:The high resolution (2.1 Å) of the current structure is impressive, and allows many new densities in the potassium transport pathway to be resolved. The authors are judicious about assigning these as potassium ions or water molecules, and explain their structural interpretations clearly. In addition to the nice structural work, the mechanistic work is thorough. A series of thoughtful experiments involving ATP hydrolysis/transport coupling under various pH and potassium concentrations bolsters the structural interpretations and lends convincing support to the mechanistic proposal. The SSME experiments are generally rigorous.Weaknesses:The present SSME experiments do not support quantitative comparisons of different mutants, as in Figures 4D and 5E. Only qualitative inferences can be drawn among different mutant constructs.

Thank you to both reviewers for your thorough review of our work. We acknowledge the limitations of SSME experiments in quantitative comparison of mutants and have revised the manuscript to address this point. In addition, we have included new ATPase data from reconstituted vesicles which we believe will help to strengthen our contention that both ATPase and transport are equally affected by Val496 mutations.

**Reviewer #2 (Recommendations for the authors):**
I have a minor editorial comment:Perhaps I am confused. However, in reference to the text in the Results: "Our WT complex displayed high levels of K+-dependent ATPase activity and generated robust transport currents (Fig. 1 - figure suppl. 1).", I do not see either K+-dependency of ATPase activity nor transport currents in Fig. 1 - figure suppl. 1. Perhaps the text needs to be edited for clarity.

Thank you for pointing this out. This confusion was caused by our removal of a panel from the revised manuscript, which depicted K+-dependent transport currents. Although this panel is somewhat redundant, given inclusion of raw SSME traces from all the mutants, it has been replaced as Fig. 1 - figure supplement 1F, thus providing a thorough characterization of the preparation used for cryo-EM analysis and supporting the statement quoted by this reviewer.

**Reviewer #3 (Recommendations for the authors):**
The authors have provided a detailed description of the SSME data collection, and followed rigorous protocols to ensure that the currents measured on a particular sensor remained stable over time.I still have reservations about the direct comparison of transport in the different mutants. Specifically, on page 6, the authors state that "The longer side chain of V496M reduces transport modestly with no effect on ATPase activity. V496R, which introduces positive charge, completely abolishes activity. V496W and V496H reduce both transport and ATPase activity by about half, perhaps due to steric hindrance for the former and partial protonation for the latter." And in figures 4D and 5B, by plotting all of the peak currents on the same graph, the authors are giving the data a quantitative veneer, when these different experiments really aren't directly comparable, especially in the absence of any controls for reconstitution efficiency.In terms of overall conclusions, for the more drastic mutant phenotypes, I think it is completely reasonable to conclude that transport is not observed. But a 2-fold difference could easily result from differences in reconstitution or sensor preparation. My suggestion would be to show example traces rather than a numeric plot in 4D/5E, to convey the qualitative nature of the mutant-to-mutant comparisons, and to re-write the text to acknowledge the shortcomings of mutant-to-mutant comparisons with SSME, and avoid commenting on the more subtle phenotypes, such as modest decreases and reductions by about half.Figure 4, supplement 1. What is S162D? I don't think it is mentioned in the main text.

We agree with the reviewer's point that quantitative comparison of different mutants by SSME is compromised by ambiguity in reconstitution. However, we do not think that display of raw SSME currents is an effective way to communicate qualitative effects to the general reader, given the complexity of these data (e.g., distinction between transient binding current seen in V496R and genuine, steady-state transport current seen in WT). So we have taken a compromise approach. To start, we have removed the transport data from the main figure (Fig. 4). Luckily, we had frozen and saved the batch of reconstituted proteoliposomes from Val496 mutants that had been used for transport assays. We therefore measured ATPase activities from these proteoliposomes - after adding a small amount of detergent to prevent buildup of electrochemical gradients (1 mg/ml decylmaltoside which is only slightly more than the critical micelle concentration of 0.87 mg/ml). Differences in ATPase activity from these proteoliposomes were very similar to those measured prior to reconstitution (i.e., data in Fig. 4d) indicating that reconstitution efficiencies were comparable for the various mutants. Furthermore, differences in SSME currents are very similar to these ATPase activities, suggesting that Val496 mutants did not affect energy coupling. These data are shown in the revised Fig. 4 - figure suppl. 1a, along with the SSME raw data and size-exclusion chromatography elution profiles (Fig. 4 - figure suppl. 1b-g). We also altered the text to point out the concern over comparing transport data from different mutants (see below). We hope that this revised presentation adequately supports the conclusion that Val496 mutations - and especially the V496R substitution - influence the passage of K+ through the tunnel without affecting mechanics of the ATP-dependent pump.

The paragraph in question now reads as follows (pg. 6-7, with additional changes to legends to Fig. 4 and Fig. 4 - figure suppl. 1):

"In order to provide experimental evidence for K+ transport through the tunnel, we made a series of substitutions to Val496 in KdpA. This residue resides near the widest part of the tunnel and is fully exposed to its interior (Fig. 4a). We made substitutions to increase its bulk (V496M and V496W) and to introduce charge (V496E, V496R and V496H). We used the AlphaFold-3 artificial intelligence structure prediction program (Jumper et al., 2021) to generate structures of these mutants and to evaluate their potential impact on tunnel dimensions. This analysis predicts that V496W and V496R reduce the radius to well below the 1.4 Å threshold required for passage of K+ or water (Fig. 4c); V496E and V496M also constrict the tunnel, but to a lesser extent. Measurements of ATPase and transport activity (Fig. 4d) show that negative charge (V496E) has no effect. The or a longer side chain of (V496M) reduces transport modestly with have no apparent effect on ATPase activity. V496R, which introduces positive charge, almost completely abolishes activity. V496W and V496H reduce both transport and ATPase activity by about half, perhaps due to steric hindrance for the former and partial protonation for the latter. Transport activity of these mutants was also measured, but quantitative comparisons are hampered by potential inconsistency in reconstitution of proteoliposomes and in preparation of sensors for SSME. To account for differences in reconstitution, we compared ATPase activity and transport currents taken from the same batch of vesicles (Fig. 4 - figure suppl. 1a). These data show that differences in ATPase activity of proteoliposomes was consistent with differences measured prior to reconstitution (Fig. 4d). Transport activity, which was derived from multiple sensors, mirrored ATPase activity, indicating that the Val496 mutants did not affect energy coupling, but simply modulated turnover rate of the pump."

S162D was included as a negative control, together with D307A. However, given the inactive mutants discussed in Fig. 5 (Asp582 and Lys586 substitutions), these seem an unnecessary distraction and have been removed from Fig. 4 - figure suppl. 1.